# Non-falciparum malaria infection and IgG seroprevalence among children under 15 years in Nigeria, 2018

Camelia Herman[1,2,12], Colleen M. Leonard [1,3,12], Perpetua Uhomoibhi[4], Mark Maire[5], Delynn Moss[6], Uwem Inyang [7], Ado Abubakar[8], Abiodun Ogunniyi[9], Nwando Mba[9], Stacie M. Greby[10], McPaul I. Okoye[10], Nnaemeka C. Iriemenam [10], Ibrahim Maikore[11], Laura Steinhardt[1] & Eric Rogier [1] ✉

*Plasmodium falciparum* (Pf) is the dominant malaria parasite in Nigeria though *P. vivax* (Pv), *P. ovale* (Po), and *P. malariae* (Pm) are also endemic. Blood samples (n = 31,234) were collected from children aged 0-14 years during a 2018 nationwide HIV survey and assayed for *Plasmodium* antigenemia, *Plasmodium* DNA, and IgG against *Plasmodium* MSP1-19 antigens. Of all children, 6.6% were estimated to have Pm infection and 1.4% Po infection with no Pv infections detected. The highest household wealth quintile was strongly protective against infection with Pm (aOR: 0.11, 95% CI: 0.05−0.22) or Po (aOR= 0.01, 0.00−0.10). Overall Pm seroprevalence was 34.2% (95% CI: 33.3-35.2) with lower estimates for Po (12.1%, 11.6-12.5) and Pv (6.3%, 6.0-6.7). Pm seropositivity was detected throughout the country with several local government areas showing >50% seroprevalence. Serological and DNA indicators show widespread exposure of Nigerian children to Pm with lower rates to Po and Pv.

As of the year 2020, Nigeria has the highest recorded malaria burden in the world, accounting for ~27% of all cases and 27% of all deaths[1]. The 2020 World Health Organization (WHO) World Malaria Report estimated 61 million *Plasmodium falciparum* cases in Nigeria with no estimates provided for the other human malarias: *P. vivax*, *P. ovale*, or *P. malariae*[1]. Prevention and treatment efforts in Nigeria between 2000 and 2019 led to a considerable decrease in malaria-associated mortality by ~55%[2]. Malaria diagnosis in Nigeria is primarily through histidine-rich protein 2 (HRP2) rapid diagnostic tests (RDTs) which are specific for *P. falciparum*, accounted for 86.3% of all diagnostic tests

provided in 2020 to patients with suspected malaria infection[1]. Furthermore, the most recent Nigerian Malaria Indicator Survey (MIS) in 2021 and Demographic Health Surveys (DHSs) in 2018 and 2021 utilized RDTs only detecting HRP2 antigen, though microscopy was also included[3].

*P. falciparum* prevalence is highest in northern and western Nigeria with holoendemic transmission occurring all year round[4]. Positive associations with increasing rainfall, proximity to bodies of water, and rural residence have been identified−all of which are indicative of more optimal *Anopheline* vector habitat[5,6]. Mass distribution

[1]Division of Parasitic Diseases and Malaria, Centers for Disease Control and Prevention, Atlanta, GA 30029, USA. [2]BeVera Solutions, Atlanta, GA 30341, USA. [3]Oak Ridge Institute for Science and Education, US. Department of Energy, Oak Ridge, TN 37831, USA. [4]National Malaria Elimination Programme, Federal Ministry of Health, Abuja, Nigeria. [5]U.S. President's Malaria Initiative, Malaria Branch, Division of Parasitic Diseases and Malaria, U.S. Centers for Disease Control and Prevention, Abuja, Nigeria. [6]Division of Foodborne, Waterborne, and Environmental Diseases, National Center for Emerging and Zoonotic Infectious Diseases, Centers for Disease Control and Prevention, Atlanta, GA 30329, USA. [7]U.S. President's Malaria Initiative, United States Agency for International Development (USAID), Abuja, Nigeria. [8]Institute of Human Virology (IHVN), Abuja, Nigeria. [9]Nigeria Centre for Disease Control (NCDC), Abuja, Nigeria. [10]Division of Global HIV and Tuberculosis, Center for Global Health, Centers for Disease Control and Prevention, Abuja, Nigeria. [11]World Health Organization, Nigeria Country Office, Abuja, Nigeria. [12]These authors contributed equally: Camelia Herman, Colleen M. Leonard. ✉e-mail: erogier@cdc.gov

of insecticide-treated bed nets (ITNs) are carried out approximately every 3 years in Nigeria, and ITNs are a primary tool for reduction of vector exposure and malaria burden in the population, though preventing mosquito bites would reduce risk for infection with any *Plasmodium* parasite. The current treatment of uncomplicated malaria in Nigeria is artemisinin combination therapy (ACT) with parenteral artesunate for severe malaria cases[7,8], and these antimalarial treatments have broad spectrum activity against non-falciparum *Plasmodium* species as well[9,10]. However, hypnozoites formed by *P. ovale* and *P. vivax* parasites are not cleared by an ACT regimen and require an 8-aminoquinoline, such as primaquine or tafenoquine for successful anti-relapse therapy[11].

Non-falciparum malaria infections in Nigeria have typically been found to have *P. falciparum* parasites mixed with another *Plasmodium* species[6,12,13], which is parallel to findings from other African countries[14]. *P. vivax* infection has recently been reported in Nigeria[15,16], adding to the growing evidence for ability of this parasite to sustain transmission cycles in human populations which are primarily Duffy antigen negative[17–20]. Infections with *P. ovale* have been reported multiple times from studies in Nigeria, but of the two subspecies of *P. ovale* (*P. ovale* curtisi and *P. ovale* wallikeri), only *P. ovale* curtisi has been identified[21,22]. When compared to other human *Plasmodium* spp., *P. malariae* typically presents as a more benign infection and infection has rarely been identified in Nigeria[13,16,22]. This current study utilizes blood samples collected during the 2018 nationwide Nigeria HIV/AIDS Indicator and Impact Survey (NAIIS) household survey to assess current infections at time of sampling and past exposure by serological data for *P. vivax*, *P. ovale*, and *P. malariae* were estimated in children under age 15 years.

## Results

### *Plasmodium* infection analyses

As part of the Nigeria Multi-disease Serologic Surveillance using Stored Specimens (NMS4) project, of 45,462 eligible children (<15 years of age) eligible for enrollment in NAIIS, dried blood spot (DBS) analysis for multiplex antigen data was able to be collected on 31,234 (68.7%) (Table 1). By 5-year age categories, the 5–9 year old category had the highest percentage of children providing a DBS with multiplex data collected (35.0%) versus the lowest percentage in the <5 year category (30.2%). Samples available for multiplex assays by sex was approximately equivalent (48.9% female), and 41.5% of children were from an urban setting in Nigeria. Of 774 local government areas (LGAs) within Nigeria, 740 (95.6%) included children's DBS samples and the median number of children enrolled per LGA was 36 (Supplementary Fig. 1).

Based on assay plate pass/fail criteria outlined in Methods and reported previously[23], 1.2% of all antigen detection assay plates failed the initial run and needed to be repeated to obtain valid antigen detection data. Of all children's samples, 38.3% of were positive for the HRP2 antigen, indicating the widespread transmission and predominance of *P. falciparum* in this population. In order to determine active *Plasmodium* infection through presence of parasite DNA, a two-part selection strategy was employed to capture infections with suspicion of non-falciparum infection as well as a separate selection for infections with a high likelihood of *P. falciparum* infection. Both selection strategies utilized the antigen data from individual DBS samples to make pragmatic selections for subsequent PCR assays from the massive available sample set. From multiplex antigen data, 613 (1.9%) samples showed an antigen profile positive for a non-HRP2 target, but with low/undetectable levels of HRP2 (Supplementary Fig. 2). In addition to these 613, an additional 600 DBS with moderate to high HRP2 levels (100 from each of the six Nigerian zones) were selected to represent *P. falciparum* infections, bringing the total number of samples selected for photo-induced electron transfer (PET)-PCR assays to 1213. Of these, 1204 (99.3%) were able to be retrieved for DNA extraction and subsequent PET-PCR speciation (Fig. 1). No

### Table 1 | Demographic characteristics for children aged <15 years with malaria laboratory data: Nigeria, 2018

| Variable | N | (%) |
|---|---|---|
| **Sex** | | |
| Male | 15,974 | (51.1) |
| Female | 15,260 | (48.9) |
| **Age group in years** | | |
| <5 | 9427 | (30.2) |
| 5–9 | 10,921 | (35.0) |
| 10–14 | 10,886 | (34.9) |
| **Wealth quintile** | | |
| Lowest | 6658 | (21.3) |
| Second | 6442 | (20.6) |
| Middle | 6511 | (20.8) |
| Fourth | 6257 | (20.0) |
| Highest | 5366 | (17.2) |
| **At least 1 ITN per 1.8 household members** | | |
| No | 26,950 | (86.3) |
| Yes | 4276 | (13.7) |
| **Place of residence** | | |
| Rural | 18,281 | (58.5) |
| Urban | 12,953 | (41.5) |
| **Total** | 31,234 | 100.0% |

*ITN* insecticide-treated net.

samples which were negative to all four *Plasmodium* antigens (*n* = 16,657, 53.3% of all) were selected for PET-PCR assays for these estimates. As a validation exercise for the selection strategy for this specific Nigerian sample set, 200 DBS were selected for PCR assays that were found to be negative to all antigen targets. Of the 200 antigen negative specimens, zero were positive for *P. ovale* or *P. vivax* DNA, three (1.5%) were positive for *P. malariae* DNA, and nine (4.5%) were positive for *P. falciparum* DNA.

From the first selection strategy (for low/no HRP2), 608 DBS were retrieved (99.2% of the 613 selected), and 164 (27.0%) of these were negative by *Plasmodium* genus primers (Fig. 1), indicating no active infection or DNA concentrations below the PET-PCR limit of detection. Of the 444 *Plasmodium* genus positive samples, single-species infections accounted for the majority with 339 (76.3%) being positive only for *P. falciparum* DNA, 13 (2.9%) positive for only *P. malariae* DNA, and four (0.9%) samples only positive for *P. ovale* DNA. Mixed infections were also common among this first selection subset with 68 (15.3% of 444 active infections) positive for both *P. falciparum* and *P. malariae* DNA, 15 (3.4%) *P. falciparum/ P. ovale* DNA positive, two (0.5%) *P. malariae/P. ovale* DNA positive, and three (0.7%) DNA positive to all three of these parasites: *P. falciparum, P. malariae, P. ovale*. From the second selection strategy (for representative *P. falciparum* infections), 596 DBS were retrieved (99.3% of the 600 selected). Of the 588 *Plasmodium* genus positive samples, all were *P. falciparum* DNA positive with 81 (13.8%) also mixed with only *P. malariae*, 15 (2.6%) mixed with only *P. ovale*, and three (0.5%) mixed infections with *P. falciparum*, *P. malariae*, and *P. ovale* DNA. None of the 1,032 *Plasmodium* DNA positive samples from the combination of the first and second selection strategies were positive for *P. vivax* DNA.

In total, 170 samples (16.5% of all *Plasmodium* PET-PCR positives) contained any *P. malariae* DNA, with 157 (92.4%) of these *P. malariae* infections mixed with either *P. falciparum* (87.6%), *P. ovale* (1.2%), or both (3.5%). In addition, 42 samples (4.1% of all *Plasmodium* PET-PCR positives) contained any *P. ovale* DNA, with 38 of these (90.5%) mixed with either *P. falciparum* (71.4%), *P. malariae* (4.8%) or both (14.3%) (Fig. 1, Supplementary Table 1). Supplementary Fig. 3 outlines the

**Fig. 1 | Number of dried blood spot samples selected for PET-PCR by selection strategy and their results: Nigeria, 2018.** Strategy 1 involved selecting DBS based on a low or negative HRP2 assay signal compared to other *Plasmodium* antigen targets. Strategy 2 selected a random sample of HRP2 positive DBS, with a target of 100 samples per Nigerian zone. Terminal boxes display number (and percentage) of each type of infection for samples that were *Plasmodium* DNA positive. DBS: dried blood spot, Pf *Plasmodium falciparum*, Pm *Plasmodium malariae*, Po *Plasmodium ovale*.

**Table 2 | Crude and adjusted odds ratio estimates for PCR-confirmed *P. malariae* or *P. ovale* infections compared to no *Plasmodium* infection with adjusted estimates considering demographic, socioeconomic, and behavioral risk factors**

| Variable | P. malariae | | P. ovale | |
|---|---|---|---|---|
| | Crude OR (95% CI) | Adjusted OR (95% CI) | Crude OR (95% CI) | Adjusted OR (95% CI) |
| **Sex** | | | | |
| Female | Ref | Ref | Ref | Ref |
| Male | 0.93 (0.69, 1.26) | 0.95 (0.70, 1.28) | 0.91 (0.49, 1.67) | 0.93 (0.50, 1.71) |
| **Age in years** | | | | |
| <5 | Ref | Ref | Ref | Ref |
| 5–9 | 1.49 (1.04, 2.16)[b] | 1.64 (1.14, 2.38)[b] | 2.57 (1.26, 6.03)[b] | 2.91 (1.43, 6.86)[b] |
| 10–14 | 1.16 (0.77, 1.73) | 1.35 (0.90, 2.03) | 1.06 (0.39, 2.75) | 1.27 (0.47, 3.31) |
| **Wealth** | | | | |
| Lowest | Ref | Ref | Ref | Ref |
| Second | 0.83 (0.56, 1.23) | 0.85 (0.58, 1.26) | 0.45 (0.19, 0.93)[b] | 0.45 (0.19, 0.94)[b] |
| Middle | 0.38 (0.24, 0.58)[b] | 0.41 (0.25, 0.64)[b] | 0.24 (0.09, 0.54)[b] | 0.25 (0.09, 0.56)[b] |
| Fourth | 0.25 (0.15, 0.40)[b] | 0.30 (0.18, 0.49)[b] | 0.17 (0.06, 0.39)[b] | 0.18 (0.06, 0.44)[b] |
| Highest | 0.09 (0.04, 0.17)[b] | 0.11 (0.05, 0.22)[b] | 0.01 (0.00, 0.09)[b] | 0.01 (0.00, 0.10)[b] |
| **Place of residence** | | | | |
| Rural | Ref | Ref | Ref | Ref |
| Urban | 0.34 (0.24, 0.47)[b] | 0.69 (0.46, 0.99)[b] | 0.32 (0.15, 0.61)[b] | 0.89 (0.39, 1.81) |
| **Mosquito net coverage[a] *(individual)*** | 1.09 (0.60, 1.87) | 1.16 (0.64, 1.98) | 0.91 (0.25, 2.68) | 1.05 (0.33, 2.73) |
| **Mosquito net coverage[a] *(community)*** | 0.81 (0.35, 1.64) | 0.75 (0.31, 1.57) | 2.39 (0.59, 7.06) | 2.62 (0.57, 8.32) |

[a]Net coverage is defined as at least 1 net per 1.8 household members.
Inclusive of *P. malariae* and *P. ovale* single-species infections, or mixed with *P. falciparum*.
[b]Statistically significant at alpha = 0.05.

distribution of all identified active infections by *Plasmodium* species and children's ages. Table 2 displays crude and adjusted estimates for odds of infection with any *P. malariae* or *P. ovale* parasites with adjusted estimates controlling for individual and community risk factors. For adjusted estimates, no association was observed between active infection and sex, but compared to children <5 years of age, children age 5–9 years had 1.64-fold increased odds of *P. malariae* infection (95% CI: 1.14–2.38), and 2.91-fold increased odds of *P. ovale* infection (95% CI: 1.43–6.86). The oldest children aged 10–14 years did not show a significant difference in odds of *P. malariae* or *P. ovale*

infection compared to the youngest children. Increasing wealth quintile showed successive decreases in odds of infection at time of sampling. Even comparing children from the middle wealth quintile to the lowest quintile showed a >50% reduction in odds of infection with either *P. malariae* (aOR: 0.41, 95% CI: 0.25–0.64) or *P. ovale* (aOR: 0.25, 0.09–0.56). Compared to the lowest wealth quintile, the highest wealth quintile had an 89% reduction in odds of active *P. malariae* infection (aOR: 0.11, 95% CI: 0.05–0.22) and 99% reduction in odds of *P. ovale* infection (aOR: 0.01, 0.00–0.10). Urban residence was found to be protective against *P. malariae* infection (aOR: 0.69, 0.46–0.99)

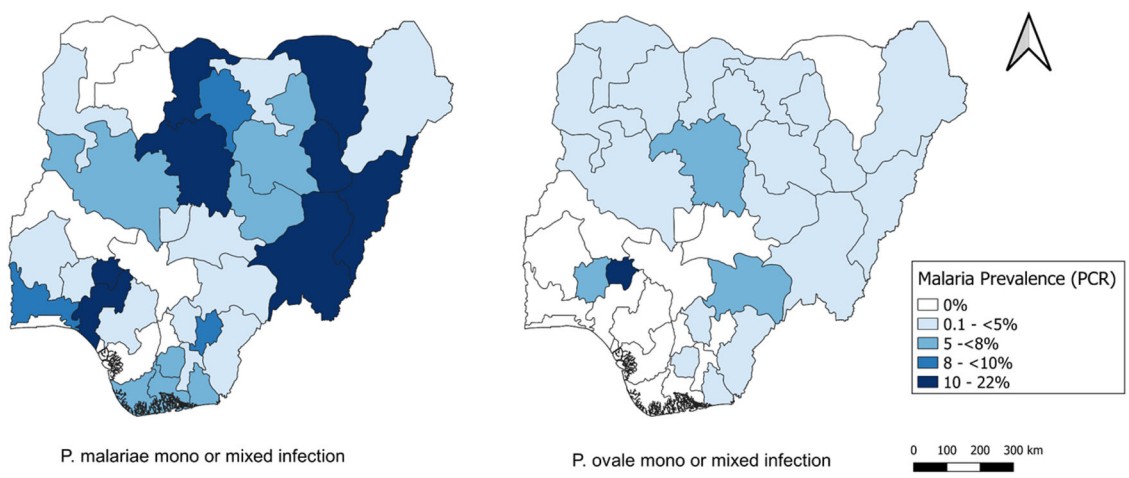

**Fig. 2 | Estimated prevalence of *P. malariae* and *P. ovale* infections by Nigerian state: 2018.** Presence of *Plasmodium* infection determined by PCR assays. Panels shown for *P. malariae* on left and *P. ovale* on right, with darker shading indicating higher estimated proportions of children infected with either of these two species.

but not *P. ovale* infection (aOR: 0.89, 0.39–1.81). Mosquito net coverage at the individual and community levels was not found to have an association with *P. malariae* or *P. ovale* infection at time of sampling. Among children with any *Plasmodium* infection, some significant associations were observed comparing infections with *P. falciparum* alone versus a mixed (or mono) infection containing *P. malariae* or *P. ovale* (or both)(Supplementary Table 2). Compared to children under 5 years of age, children aged 5–9 years were over twice as likely (aOR: 2.24, 1.07–5.41) to have an infection containing *P. ovale* versus an infection with only *P. falciparum*. Increasing wealth status had a supplemental protective effect against *P. malariae* or *P. ovale* infections versus *P. falciparum*-only infections, but this was only significant for the middle wealth quintile (for both *P. malariae* and *P. ovale*) and the highest quintile (for only *P. ovale*).

By extrapolating the proportion of PET-PCR confirmed *P. malariae* and *P. ovale* infections (single-species or mixed) obtained through the two selection strategies, 6.6% of all children enrolled were estimated to have *P. malariae* mono or mixed-infection, and 1.4% *P. ovale* mono or mixed infection. As no active *P. vivax* infections were identified by PET-PCR, no state or national estimates were available for that species. State-level estimates for all 36 Nigerian states plus federal capital territory (FCT) and are displayed in Fig. 2. By state, estimated *P. malariae* prevalence in all children age <15 years ranged from 0.0 to 22.0%, and *P. ovale* prevalence ranged from 0.0 to 11.9%. For both *Plasmodium* species, infection prevalence was spread throughout the country in a heterogeneous manner.

### Serological analysis
A total of 31,234 DBS from children age <15 years were assessed for IgG to the MSP1-19kD (MSP1) antigens from all four human *Plasmodium* spp. Data from 225 DBS (0.72% of all) with MFI-bg reads >500 for the glutathione-*S*-transferase (GST) internal control were excluded from IgG analysis for quality assurance as explained in Methods. No correlation in IgG binding was observed in comparing the assay signals among all four merozoite surface protein 1 (MSP1) orthologous antigens (Supplementary Fig. 4). A two-component finite mixture model (FMM) was able to identify the putative seronegative children in the study population by the IgG assay signals for all three non-falciparum MSP1 antigens (Supplementary Fig. 5). Serological evidence for *P. malariae* exposure found 34.2% (95% CI: 33.3–35.2) of all children with anti-PmMSP1 IgG, with state seroprevalence ranging from 5.2 to 57.6% (Supplementary Table 3). *P. ovale* exposure by anti-PoMSP1 IgG seropositivity was 12.1% (95% CI: 11.6–12.5) for children overall with state seroprevalence estimates ranging from 2.9 to 22.3%. *P. vivax*

exposure by anti-PvMSP1 IgG seropositivity was lowest at 6.3% (95% CI: 6.0–6.7) for all children, with state ranges from 1.7 to 17.6%. Seroprevalence by LGA for these three antigens is displayed in Fig. 3 with IgG seroprevalence to PfMSP1 included for illustrative comparison. Relative standard errors (RSEs) for seroprevalence by LGA are displayed in Supplemental Fig. 6.

Indicating a greater amount of cumulative malaria exposure as Nigerian children age, increasing IgG levels were observed for all three non-falciparum MSP1 antigens with increasing age, though most accentuated for PmMSP1 (Fig. 4). To model the relationship between age and IgG seroprevalence, reversible catalytic conversion models were utilized to estimate the rate IgG acquisition to these three non-falciparum antigens (Fig. 5). The seroconversion rate (SCR) was significantly different among all three antigens: PmMSP1 at 0.096 (95% CI: 0.091–0.102), PoMSP1 at 0.034 (95% CI: 0.030–0.038), and PvMSP1 at 0.015 (95% CI: 0.014–0.017) (Fig. 5a). Though no difference in SCR or seroprevalence was observed between sexes (Supplementary Fig. 7), rural residence and lower socioeconomic status (SES) were both strong indicators of higher rates of non-falciparum exposure as children aged. Children residing in rural settings showed SCRs for PmMSP1 (0.116), PoMSP1 (0.041), and PvMSP1 (0.018) which were significantly higher than SCR estimates for children living in urban settings: 0.055, 0.018, and 0.009, respectively (Fig. 5b). Higher SES by increasing wealth quintile provided a dose-response protective effect against non-falciparum exposure as children aged, with children in the highest wealth quintile having the lowest SCR and seroprevalence estimates for all three antigens. The highest estimates for non-falciparum exposure by SCR and seroprevalence for all three non-falciparum antigens were among children in the lowest wealth quintile (Fig. 5c). For the two strongest protective effects against MSP1 seropositivity, urban residence and dwelling having air conditioning, children living in conditions with both of these protective factors showed the slowest acquisition of anti-MSP1 IgG with age, whereas children with neither of these protective factors showed the fastest acquisition of these antibodies (Supplementary Fig. 8).

Reflecting results from the seroconversion analyses, unadjusted and multivariate logistic regression to investigate individual or demographic associations with non-falciparum MSP1 seropositivity found increasing age, rural residence, and decreasing SES to all be strong indicators of increased non-falciparum malaria exposure (Table 3). Versus rural residence, urban residence showed significant reduction in adjusted odds of seropositivity to PmMSP1 by 58% (aOR 95% CI: 0.37–0.46), reduction to PoMSP1 by 47% (aOR 95% CI: 0.47–0.59), and reduction to PvMSP1 by 32% (aOR 95% CI: 0.59–0.79).

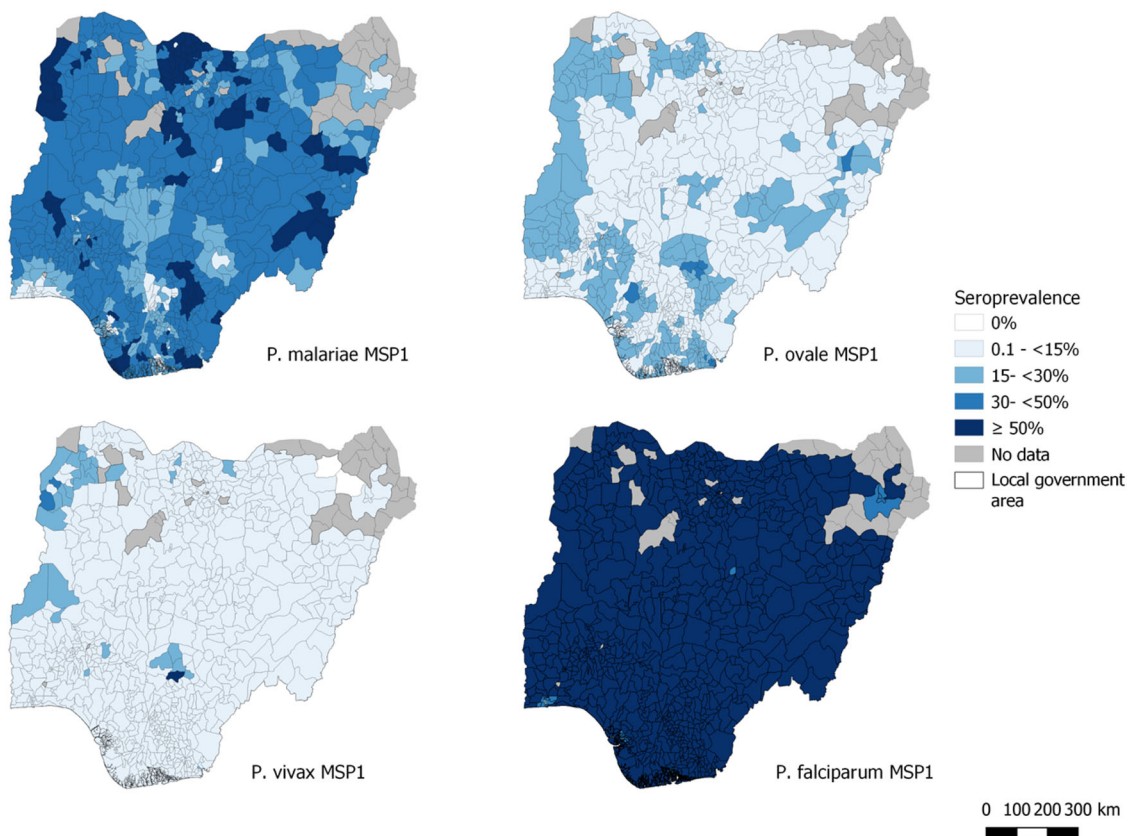

**Fig. 3 | Seroprevalence to *P. malariae*, *P. ovale*, *P. vivax* and *P. falciparum* MSP1 antigens among children under 15 years: Nigeria, 2018.** Darker shading indicates higher seroprevalence by local government area (LGA) with gray areas showing LGAs without samples available for IgG data collection. MSP1: merozoite surface protein 1, 19kD region.

When compared to the lowest wealth quintile, children in the highest wealth quintile had a 63% reduction in odds of PmMSP1 seropositivity (aOR 95% CI: 0.23−0.30), 48% reduction in odds of PoMSP1 seropositivity (aOR 95% CI: 0.44−0.62), and 71% reduction in odds of PvMSP1 seropositivity (aOR 95% CI: 0.23−0.36). Similar to the risk factor analysis for active infections, reductions in adjusted odds were observed even in comparing the middle to the lowest wealth quintile, though this reduction did not reach statistical significance for PmMSP1 and PvMSP1. Unadjusted bivariate associations with non-falciparum MSP1 seropositivity found consistent positive associations with increasing number of members dwelling in household and owning livestock, whereas living in a house with finished roofing, finished walls, and electricity/air conditioning were all negatively associated with MSP1 seropositivity (Supplementary Table 4).

**Combination of PET-PCR and serological data**
Non-significant increases were seen in median PmMSP1, PoMSP1, and PvMSP1 IgG levels for children actively infected with *P. falciparum* alone (Supplementary Fig. 9), further alluding to the inability of IgG developed against *P. falciparum* antigens to bind the non-falciparum MSP1 antigens. Further emphasizing the connection between parasite exposure and IgG responses, substantial and significant increases in PmMSP1 IgG levels were seen for all age groups when actively infected with *P. malariae* (Supplementary Fig. 10A), but no increases in PoMSP1 IgG levels for these same children (Supplementary Fig. 10C). Furthermore, increases were observed in PoMSP1 IgG levels for children in all age groups actively infected with *P. ovale* (Supplementary Fig. 10B), but not for PmMSP1 IgG levels in these same children (Supplementary Fig. 10D). For each of the 36 Nigerian states and FCT, *P. malariae* and *P. ovale* infection prevalence estimates were plotted against SCR estimates from the same state. Figure 6 shows a positive, linear trend

among these two estimates for *P. malariae* ($R^2 = 0.23$), but no observable trend for *P. ovale* ($R^2 = 0.00$). The trend between *P. malariae* infection and exposure was more clearly elucidated as more infections and seropositive children were observed, and seroconversion estimates were able to be modelled for 86.5% (32/37) of states. However, for the lower prevalence of infections and observed seropositivity, the serocatalytic model for PoMSP1 converged for only 29.7% (11/37) of states, and this state-level correlation between infections and exposure by seropositivity was not observed.

## Discussion
This current study from the NAIIS 2018 investigated exposure of Nigerian children aged <15 years to non-falciparum malaria by detecting infections through antigen and DNA presence and estimating previous exposure by anti-malarial IgG presence. The findings presented here estimated that 6.6% of Nigerian children enrolled in NAIIS 2018 were infected with *P. malariae* at the time of sampling and 1.4% infected with *P. ovale*. However, mono-infections with either of these two species were rare, and the majority of detected *P. malariae* infections (91.2%) and *P. ovale* infections (85.7%) were mixed with *P. falciparum* parasites. A report outlining comprehensive analyses for *P. falciparum* infection and exposure among persons of all ages from NAIIS 2018 is forthcoming. Though *P. falciparum* is well-documented to be the dominant human malaria species in Nigeria[1,3], reports of human exposure to the other three human malarias have also been produced over the past decades. Advances in PCR technologies have made sensitive detection of nucleic acids of these parasites more pragmatic, and IgG-detecting serological techniques can serve as a sensitive proxy for previous infection even after parasites have cleared from the host[24]. Though the clinical relevance of *P. falciparum* is predominant in Nigeria, it is evident from this current study that

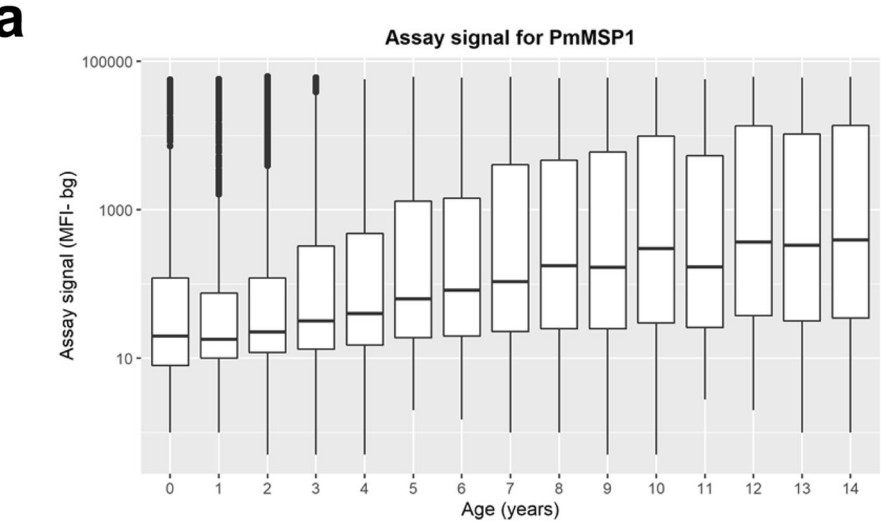

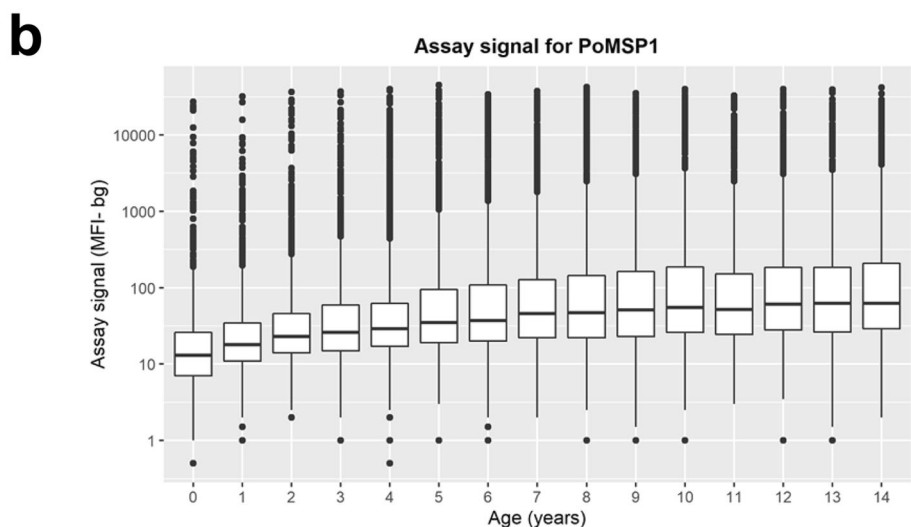

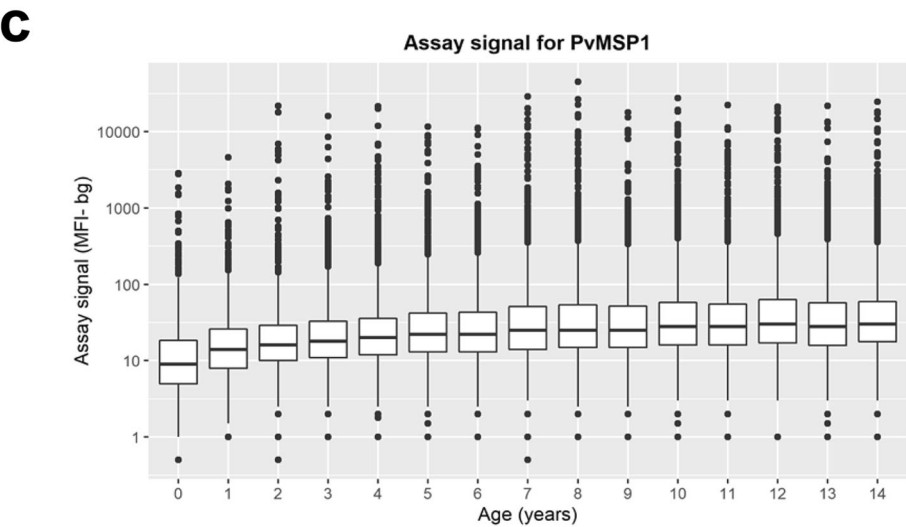

**Fig. 4 | Changes in IgG antibody levels in Nigerian children during the first 14 years of life.** IgG antibody levels against PmMSP1 (**a**), PoMSP1 (**b**), and PvMSP1 (**c**) antigens by age. Boxes display interquartile range (IQR), with horizontal lines as median assay signals and whiskers extending 1.5× IQR above and below boxes and points at >1.5× IQR. MFI-bg: median fluorescence intensity minus background assay signal. For all plots, $n = 31,234$ biologically independent samples.

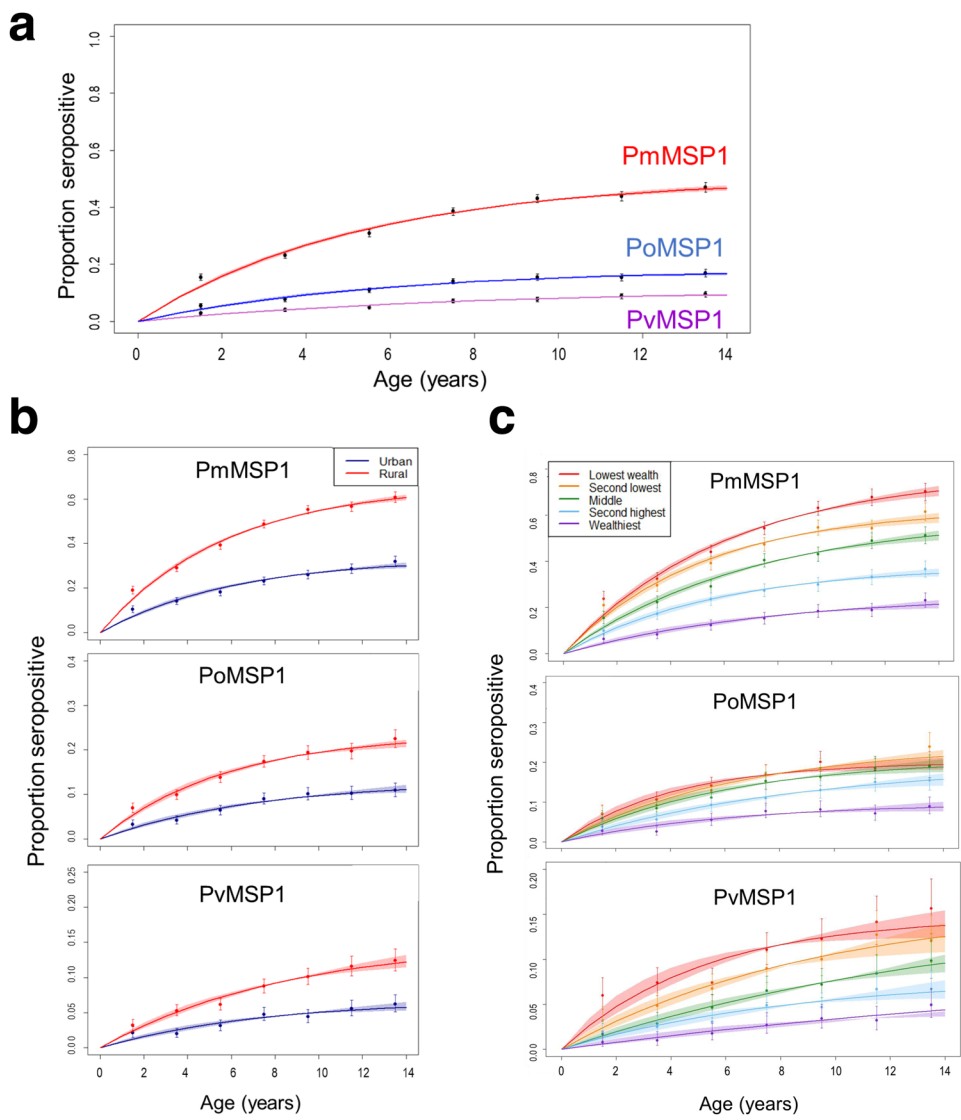

**Fig. 5 | Seropositivity by age for IgG against PmMSP1, PoMSP1, and PvMSP1 antigens among children under 15 years: Nigeria, 2018.** All participants (**a**), and as separated by urban or rural residence (**b**), or wealth quintile (**c**). Dots represent the proportion seropositive and error bars represent the 95% confidence intervals for seropositivity for each age group. Curves represent the fit of a catalytic ser-oconversion model, and shaded areas represent the 95% credible intervals of the model predictions. For all plots, $n = 31,234$ biologically independent samples.

significant non-falciparum malaria parasite burden is also occurring—at least in the youngest segment of this population. Even as able to cause severe disease, *P. malariae* and *P. ovale* parasites typically present as a more benign malaria infection[25], and since most of these infections were mixed with *P. falciparum*, it is likely antimalarial treatment for *P. falciparum* malaria will also clear these other parasites. In human populations largely negative for the Duffy red blood cell antigen, such as in Nigeria[16,26], *P. vivax* infections are now well-documented to occur in some populations, though at very low prevalence rates[20]. Overall findings from this current study emphasize the heterogeneous nature of non-falciparum species distribution in Nigeria. While *P. falciparum* transmission appears to follow a strong north-south gradient with higher transmission in the northern states[1,3], this finding was not observed for children's exposure to the non-falciparum species, and no discernable geospatial pattern was noted in this current Nigeria study.

Previous MISs (utilizing participant enrollment at households), as well as studies in limited areas of the country, had observed similar trends with *P. malariae* infection prevalence higher than *P. ovale* and most infections with either of these two species also containing

*P. falciparum*. A survey of two towns in southwest Nigeria enrolling non-treatment-seeking persons and detecting infections by PCRs found 11.7% of participants with *P. malariae* infections and 6.8% with *P. ovale* infections, but >97% of these infections also contained *P. falciparum*[27]. Another recent study of asymptomatic persons age <20 years in southwestern Nigeria found 34.1% of persons with *P. malariae* infection, and 15.7% with *P. ovale*, with 81.5% of these infections including *P. falciparum* parasites[6]. Health facility surveys of treatment-seeking persons in Nigeria have observed similar findings with *P. malariae* as the most frequently identified non-falciparum parasite followed by *P. ovale*[12,16,28]. However, since nearly all of these infections include *P. falciparum*, it is unclear how much these non-falciparum species contribute to clinical disease in Nigeria. Though active *P. vivax* infections were not observed in this current study, recent reports have identified endemic[15,16], and imported[29], *P. vivax* infections from Nigeria and provided evidence for this parasite being present in the country.

The trend of higher *P. malariae* and *P. ovale* prevalence among children age 5–9 years compared to other children as observed in this current study has also been seen in other countries in sub-Saharan

**Table 3 | Crude and adjusted odds ratio estimates for risk factors for IgG seropositivity to PmMSP1, PoMSP1, and PvMSP1 antigens by demographic, socioeconomic, and behavioral risk factors**

| Variable | P. malariae MSP1 | | P. ovale MSP1 | | P. vivax MSP1 | |
|---|---|---|---|---|---|---|
| | Crude OR (95% CI) | Adjusted OR (95% CI) | Crude OR (95% CI) | Adjusted OR (95% CI) | Crude OR (95% CI) | Adjusted OR (95% CI) |
| **Sex** | | | | | | |
| Female | Ref | Ref | Ref | Ref | Ref | Ref |
| Male | 0.97 (0.92, 1.02) | 0.97 (0.92, 1.03) | 0.94 (0.87, 1.01) | 0.94 (0.87, 1.01) | 0.96 (0.87, 1.06) | 0.96 (0.87, 1.05) |
| **Age in years** | | | | | | |
| <5 | Ref | Ref | Ref | Ref | Ref | Ref |
| 5–9 | 2.83 (2.64, 3.04)[b] | 2.80 (2.61, 3.01)[b] | 2.25 (2.03, 2.49)[b] | 2.25 (2.03, 2.49)[b] | 1.94 (1.69, 2.22)[b] | 1.94 (1.69, 2.23)[b] |
| 10–14 | 5.03 (4.66, 5.43)[b] | 5.12 (4.74, 5.53)[b] | 3.04 (2.74, 3.38)[b] | 3.13 (2.82, 3.47)[b] | 3.02 (2.63, 3.46)[b] | 3.15 (2.75, 3.61)[b] |
| **Wealth quintile** | | | | | | |
| Lowest | Ref | Ref | Ref | Ref | Ref | Ref |
| Second | 0.79 (0.73, 0.87)[b] | 0.83 (0.76, 0.91)[b] | 0.97 (0.87, 1.09) | 1.01 (0.91, 1.14) | 0.83 (0.73, 0.95)[b] | 0.84 (0.73, 0.97)[b] |
| Middle | 0.58 (0.52, 0.64)[b] | 0.64 (0.58, 0.72)[b] | 0.86 (0.76, 0.97)[b] | 0.96 (0.85, 1.09) | 0.58 (0.50, 0.67)[b] | 0.60 (0.52, 0.71)[b] |
| Fourth | 0.35 (0.32, 0.39)[b] | 0.44 (0.40, 0.50)[b] | 0.63 (0.55, 0.71)[b] | 0.79 (0.69, 0.91)[b] | 0.41 (0.35, 0.48)[b] | 0.46 (0.39, 0.55)[b] |
| Highest | 0.19 (0.16, 0.21)[b] | 0.27 (0.23, 0.30)[b] | 0.36 (0.31, 0.42)[b] | 0.52 (0.44, 0.62)[b] | 0.24 (0.19, 0.29)[b] | 0.29 (0.23, 0.36)[b] |
| **Place of residence** | | | | | | |
| Rural | Ref | Ref | Ref | Ref | Ref | Ref |
| Urban | 0.28 (0.26, 0.31)[b] | 0.42 (0.37, 0.46)[b] | 0.44 (0.40, 0.48)[b] | 0.53 (0.47, 0.59)[b] | 0.46 (0.40, 0.52)[b] | 0.68 (0.59, 0.79)[b] |
| **Mosquito net coverage[a] (individual)** | 1.05 (0.95, 1.15) | 1.09 (0.99, 1.20) | 0.98 (0.86, 1.12) | 1.00 (0.88, 1.14) | 1.02 (0.86, 1.21) | 1.05 (0.89, 1.25) |
| **Mosquito net coverage[a] (community)** | 1.49 (1.22, 1.82)[b] | 1.33 (1.10, 1.60)[b] | 1.26 (1.03, 1.54)[b] | 1.18 (0.97, 1.43) | 1.13 (0.87, 1.48) | 1.04 (0.80, 1.35) |

[a]Net coverage is defined as at least 1 net per 1.8 household members.
[b]Statistically significant.

Africa[30,31], and reflects different *P. falciparum* infection rates seen among children in some studies[32–34]. Even when considering *P. falciparum* mono-infections as the referent, *P. ovale* infections were still significantly higher in this age category. It is unclear why children in this age group would be more susceptible to malaria infection, but it may reflect behavioral changes (including outdoor activities and net usage[34]) and a lack of acquired immunity which would occur through multiple prior exposures. Odds of *P. malariae* or *P. ovale* infection at the time of study enrollment were lower for children living in urban areas, though this was only statistically significant for *P. malariae*. This finding has been consistently observed for studies investigating *P. falciparum* prevalence and is likely a factor of improved housing structures and less-optimal Anopheline habitat[35]. The most striking association with reduced odds of *P. malariae* and *P. ovale* infections was observed with increasing SES as measured by wealth quintiles. When modeling controlled for other covariates, a consistent negative trend for infection prevalence was seen with increasing wealth quintile with the highest quintile showing an 89% reduction in odds of *P. malariae* infection and 99% reduction in odds of *P. ovale* infection compared with children included in the lowest quintile. Even children in the second lowest wealth quintile showed lower odds of *P. malariae* (reduced by 15%) and *P. ovale* (reduced by 55%) when compared to children in the lowest SES category. These same prominent associations with malaria infection and wealth quintile had also been observed by the 2010 and 2015 Nigerian MISs, though asymptomatic parasite prevalence was only reported for *P. falciparum*[3]. In addition, infections with these two non-falciparum species had also been seen to be higher among pregnant woman of lower SES in western Africa[36]. These consistent protective associations with higher SES are multifactorial with access to better nutrition, improved housing, and a variety of other aspects afforded by those with higher SES. Known protective factors against vector exposure would inherently lead to protection against establishment of *Plasmodium* infection. Indeed, in our bivariate analyses, correlates of wealth (finished walls, finished roofing, electricity,

air conditioning) were all significantly protective against exposure to all three non-falciparum malaria parasites.

*Plasmodium* exposure (at time of sampling or previously in life) to non-*falciparum* species through IgG serology found several LGAs throughout the country with >50% *P. malariae* seropositivity rate, with much lower estimates for IgG against *P. ovale* or *P. vivax*. In utilizing the *Plasmodium* spp. MSP1-19kD IgG targets in this study, previous data had shown the limited degree of IgG cross-binding among these antigens[37], and we did not observe any significant correlation in the magnitude of IgG binding among these antigens in this Nigerian population (Supplementary Fig. 4). This serological data coupled with the active infection data provides strong evidence for *P. malariae* as the second most common malaria species in Nigeria behind *P. falciparum*. Estimated *P. ovale* seropositivity showed rates approaching 50% in some LGAs, but similar to *P. malariae*, these areas of exposure were dispersed throughout the country. *P. vivax* seropositivity varied between 0.1 and 15% for the vast majority of LGAs, though only two of 740 LGAs (0.3%) had a complete absence of PvMSP1 seropositive children. Seroprevalence estimates by individual LGAs where the RSEs are >30% should be interpreted with caution. Specifically, the overall lower IgG seroprevalence to the *P. ovale* and *P. vivax* MSP1 targets led to higher RSEs by LGA with most exceeding an RSE of 30%. Overall, this study estimates that there is moderate transmission of *P. malariae* occurring throughout Nigeria, and by age 15 years, children have approximately a 40% likelihood of have been infected at some point in their lives. The SCR estimate for PmMSP1 was also significantly higher than PoMSP1 and PvMSP1, and estimated by year of life, a child in Nigeria has a 9.6% likelihood of seroconverting to PmMSP1, but much lower likelihood for PoMSP1 (3.4%) or PvMSP1 (1.5%). A previous study in two states in southern Nigeria enrolling persons of all ages found high *P. malariae* seroprevalence of >70% for each of the five *P. malariae* antigens tested[12], and in neighboring Benin, high seroprevalence to PmMSP1 (68%) and PoMSP1 (57%) was detected among microscopy-negative adults[38]. As this current study only sampled children <15 y old,

**a**

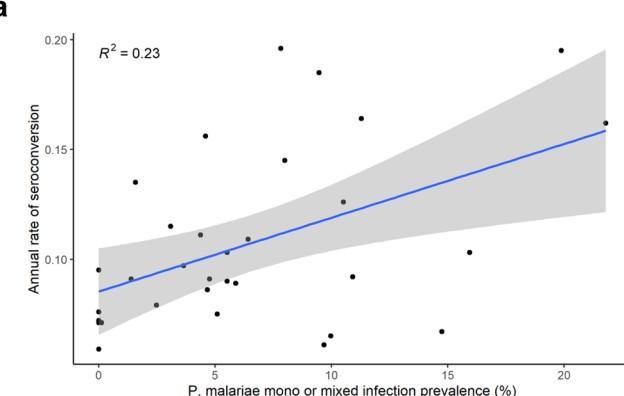

**b**

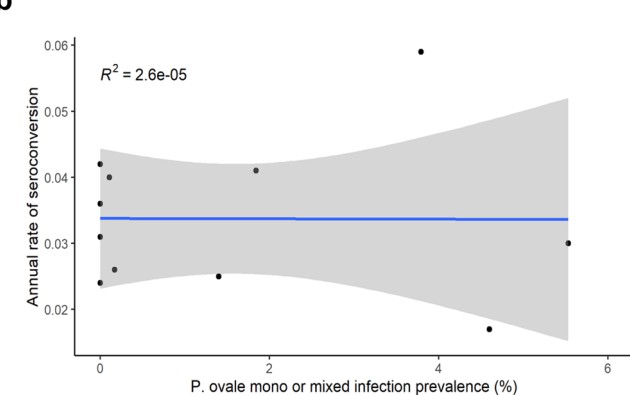

**Fig. 6 | Correlation of active infection and serological data for both *P. malariae* and *P. ovale*: Nigeria, 2018.** Association between infection prevalence of *P. malariae* (**a**) and *P. ovale* (**b**) infections per state and estimated annual ser-oconversion rate (SCR) by state to PmMSP1 and PoMSP1, respectively. Each point represents one state, blue line is the linear regression line of best fit, shading indicates the standard error, and inset text the $R^2$ estimate. For PmMSP1, the ser-ocatalytic model converged for 86.5% (32/37) of states to provide a seroconversion rate, and for PoMSP1, the model converged for 29.7% (11/37) of states.

the seropositivity estimates and SCRs for *P. malariae* and *P. ovale* are lower; though the same trend of higher *P. malariae* seroprevalence compared to *P. ovale* and *P. vivax* was observed.

Increased household wealth and residing in an urban area were found to have a protective effect against lifetime exposure to all three non-falciparum species, and this finding was consistent with the PET-PCR active infection estimates from this study. Due to greater numbers of seropositive children (versus children with active infections), greater statistical power was allowed by the IgG data to assess non-falciparum exposure and include estimates for *P. vivax*. Lifetime exposure to all three non-falciparum malarias trended significantly lower with increasing SES, and this significant reduction in odds was seen even in comparing the second wealth quintile to the lowest wealth quintile for both *P. malariae* and *P. vivax* exposure. Urban residence was found to be a strong protective factor against exposure to all three non-falciparum species, and this protection appears to be initiated very early in life as PmMSP1, PoMSP1, and PvMSP1 seroprevalence curves are completely separated by urban/rural residence by 1 year of age. Similarly, separation of ser-oprevalence curves by SES quintiles was observed by approximately 4 years of age. Individual mosquito net coverage was not found to have a protective effect against *P. malariae*, *P. ovale* or *P. vivax* infection or exposure. The community net coverage indicator showed a positive association with having antibodies against *P. malariae* MSP1 antigens, possibly showing the connection between higher malaria transmission and greater awareness of needing to

sleep under a net. However, the net coverage associations should be interpreted cautiously as this variable only indicates whether there was at least 1 net per 1.8 persons living in the household and does not indicate the actual frequency of net usage for a child.

This study has several important limitations. The cross-sectional design and timeframe of the study (during the rainy season) prevents assessing the effects of temporality, such as seasonality on active infection. The household NAIIS sampling design captured people more likely to be asymptomatic or mildly symptomatic, so the epidemiology of malaria species among persons with symptomatic malaria was unable to be assessed. In addition, very low density infections for these non-falciparum infections would likely be missed by antigen detection and/or PCR assays, so the absolute estimates for prevalence of infections with these species are certainly an underestimate of the true prevalence. Though sampling design was based on Nigerian census data, the NAIIS survey was initially designed to generate state and national estimates of HIV prevalence. Also, the number of samples tested by PCR ($n = 1204$) was too small to permit generation of representative infection prevalence estimates by LGA.

The lack of routine microscopy and the ubiquitous use of HRP2-based RDTs often preclude detecting non-falciparum infections in clinics and communities within Nigeria. While this study and other previously published reports have found the majority of non-falciparum infections were mixed infections with *P. falciparum*, the distribution and prevalence of non-falciparum species may shift as malaria declines in a region[39]. Laboratory analyses of children's blood samples from a nationwide 2018 household survey found *P. malariae* and *P. ovale* infections across Nigeria in children enrolled at households, with serological evidence for exposure to all three non-falciparum human malarias throughout the country. Though at a lower risk for severe disease and likely not affecting clinical diagnostic algorithms, these non-falciparum species can still lead to population morbidity through chronic anemia, acute febrile illness, pregnancy complications, and other factors. Accurate epidemiology of non-falciparum malaria will be needed as *P. falciparum* burden declines in Nigeria and, data on all *Plasmodium* species will need to be considered as the country moves toward malaria elimination in the future.

## Methods
### Ethics

All participants provided informed consent/assent before enrollment into the NAIIS survey. For children age <10 years, consent for biomarker testing and consent to future testing was granted by parent or guardian; assent was received from children age 10–14 years. Secondary laboratory testing for malaria biomarkers as part of the Nigeria Multi-disease Serologic Surveillance using Stored Specimens (NMS4) project was approved by the National Health Research Ethics Committee of Nigeria (NHREC/01/01/2007) and determined to not involve human research by the Centers for Disease Control and Prevention Human Subjects office (project 0900f3eb819d4c63).

### Survey design

NAIIS 2018 was a national HIV population-based household survey led by the Government of Nigeria under the Federal Ministry of Health, and survey methods have been previously reported[40]. The sampling design utilized a stratified two-stage cluster sample with 37 strata (the 36 states and Federal Capital Territory, FCT). Enumeration areas (EAs) were selected within each state in the first stage with probabilities proportionate to estimated population size according to the projected 2018 number of households (based on the 2006 census data). When calculating the number of EAs to be selected per state, state variability of household size was considered. This sampling frame produced 662,855 EAs, of which 4035 were selected for NAIIS. For the second stage, a household listing exercise was carried out in all selected EAs, and a random sample of 28 households per EA were chosen for

inclusion. The resulting sample included 101,580 households to be approached for enrollment of household participants into NAIIS.

## Laboratory data collection for malaria biomarkers

This report from the NMS4 project is inclusive of data from all children (persons <15 years of age) who provided a DBS for the NAIIS survey. Priority was given this this subset of samples as children represent the segment of the population most susceptible to adverse outcomes from malaria infection. Collection of data for persons >15 years of age is still ongoing, and an upcoming report will outline those findings. All multiplex antigen and IgG detection assays were performed at the National Reference Laboratory (NRL) of the Nigeria Centre for Disease Control (NCDC) in Gaduwa, Nigeria; DNA assays were performed at at the US Centers for Disease Control and Prevention in Atlanta, GA. Multiplex antigen and IgG detection were conducted for all available DBS specimens for children <15 years of age, and DNA assays performed for only samples selected as described below.

## Multiplex bead-based *Plasmodium* antigen detection assay

To rehydrate blood samples from DBS, a 6 mm punch (-10 μL whole blood) was taken from each DBS sample and blood eluted to a 1:40 concentration in blocking buffer (Buffer B: Phosphate Buffered Saline (PBS) pH 7.2, 0.5% Bovine Serum Albumin (BSA), 0.05% Tween 20, 0.02% sodium azide, 0.5% polyvinyl alcohol, 0.8% polyvinylpyrrolidone and 3 μg/mL *Escherichia coli* extract)[41]. Elutions were stored at 4 °C until testing. As described previously[23], samples were assayed in singlet at the 1:40 whole blood dilution for histidine-rich protein 2 (HRP2), pan-*Plasmodium* aldolase (pAldolase), pan-*Plasmodium* lactate dehydrogenase (pLDH), and *P. vivax* LDH (PvLDH). Assay plates were read on MAGPIX™ instruments (Luminex Corp, Austin, TX), and with a target of at least 50 beads/region, the median fluorescence intensity (MFI) was generated for each analyte by Luminex xPonent® software version 4.2 (Luminex Corp). For each assay plate, the MFI signal for sample dilution buffer (buffer background) was subtracted for each target's MFI signal to provide a MFI-bg assay signal for analysis. Plate-to-plate variation was accounted for by assessment of controls on each plate, inclusive of antigen negative blood (at 1:40 dilution) and a 4-point titration curve of a recombinant antigen positive pool. As described previously[23], assay plates 'failed' if the negative controls showed a positive signal, or if the positive controls for both pLDH and HRP2 were both 2 standard deviations outside of the cumulative moving average of the MFI-bg signal for assay plates. Previous analysis found 1.2% of initial antigen detection plates failed[23], and failed plates were re-run to obtain valid data. To determine the assay signal threshold for antigen positivity, a finite mixture model approach was employed to estimate the 'antigen negative' population within these data and then establishing a cutoff using the mean + 2 standard deviations from this distribution[42].

## Multiplex bead-based IgG detection assay

The IgG-detecting multiplex bead assay (MBA) was performed as described previously[41], and included targets for a variety of infectious diseases, as well as vaccine-preventable disease targets. Assay plates were run on same machines utilizing the same software as the antigen detection assay. From the DBS elution described above, an approximate final serum dilution of 1:400 was used for the assay with samples run in singlet. Analysis in this current study was restricted to the *Schistosoma japonicum* glutathione-*S*-transferase (GST) internal control antigen and the four *Plasmodium* merozoite surface protein 1, 19kD (MSP1) antigens from the human malarias: *P. falciparum*, *P. malariae*, *P.* ovale, and *P. vivax*[37]. Plate-to-plate variation was accounted for by assessment of controls included on each assay plate, and a similar 'pass/fail' structure as listed above for the multiplex antigen assay. Assay controls were inclusive of a 1:400 dilution of a sera pool from U.S. residents (no malaria exposure), and a positive control

pool of persons residing in various *P. falciparum* transmission settings. In a similar manner as described previously[43], Levey-Jennings charts were generated for the positive control to track deviations in assay signals over time, and an assay plate 'failed' if negative controls showed a positive signal, or positive controls were 2 standard deviations outside of the cumulative moving average. Of 523 IgG assay plates, 31 (0.6%) failed and needed to be repeated to obtain valid data.

## Sample selection for DNA assays

A two-part selection strategy was utilized based on the *Plasmodium* antigen data to select samples for further DNA assays, and to provide a representative estimate of the burden of non-*falciparum* malaria in Nigeria. Results from both sampling designs for DNA testing were considered when presenting national and state-level active infection estimates. Estimates are presented either as non-falciparum single-species infection, or a *P. falciparum* infection mixed with another species, or as a combination of these two (i.e., 'any *P. ovale* present', etc.).

As described in previous studies[44–46], and illustrated in Supplementary Fig. 2, the first set of samples was selected based on the ratio of HRP2 antigen to the other three *Plasmodium* targets. If any specimen was positive to a non-HRP2 target, but found to have a low or negative HRP2 signal relative to other *Plasmodium* antigens by visual identification, this sample was selected for suspicion of non-*P. falciparum* malaria infection (or potentially *P. falciparum* deleting either *pfhrp2* and/or *pfhrp3* genes)[44,46]. This selection strategy would favor the identification of single species non-*P. falciparum* infections, or infections where *P. falciparum* was not the dominant *Plasmodium* species present.

The second selection criteria involved specimens positive to HRP2, which would indicate active *P. falciparum* infection. For each of the six zones in Nigeria (North West, North East, North Central, South West, South East, South South) a range of HRP2 levels (100–1000 ng/mL) was chosen from DBS from children in each zone, with 100 DBS randomly selected to geographically represent *P. falciparum* infections for each zone (target sample size of 600), and ultimately the country as a whole. This selection strategy allowed assessment by DNA assays of *P. falciparum* infections that may also harbor another *Plasmodium* species (mixed infections). As only *P. falciparum* infections were selected from the second selection strategy, the percentages of mixed infections were calculated for each Nigerian state and extrapolated to the total number of samples with antigen data from that Nigerian state.

## Photo-induced electron transfer (PET) PCR assays

One 6 mm DBS punch was placed into a 1.5 mL tube for processing according to manufacturer's instructions. The QIAamp DNA blood mini kit (Qiagen, Valencia, CA, USA) was used to elute whole DNA in 150 μL of elution buffer, aliquoted, and stored at −20 °C until use. The PET-PCR reaction was performed with primer targets for the 18 S ribosomal RNA gene for *Plasmodium* genus and for *P. falciparum* as described previously[47]. PET-PCR primer targets for the *P. ovale* reticulocyte binding protein 2 (*rbp2*) gene have been published previously[48], as well as the PET-PCR dihydrofolate reductase-thymidylate synthase (*dhfr-ts*) gene for *P. malariae* and *P. vivax*[49]. Primer sequences for all targets listed above are shown in Supplementary Data 1. Reactions were performed in a 20 μL reaction containing 2X TaqMan Environmental buffer 2.0 (Applied BioSystems, Grand Island, NY, USA), 125 nM each of forward and reverse primers except for the *P. falciparum* HEX-labelled primer which was used at a 62.5 nM. Each sample was run with 2 μL of DNA in the PCR reaction with the following cycling parameters: initial hot-start at 95 °C for 10 min, followed by 45 cycles of denaturation at 95 °C for 10 s, annealing at 60 °C for 40 s. Fluorescence channels were selected to detect each fluorescently labelled primer-set and cycle threshold (CT) values recorded at the end

of each annealing step. All assays were performed using Agilent Mx3005pro thermocyclers (Agilent technologies, Santa Clara, CA, USA) with Agilent Aria 2.0 analytical software. The PET-PCR criteria for DNA positivity for any primer set was a CT value < 40.0.

## Statistical analysis

Survey questionnaire data cleaning was conducted using Census and Survey Processing System (CSPro) version 7.7.2 (U.S. Census Bureau; census.gov/data/software/cspro) and SAS version 9.4 (SAS Institute, Cary, NC). For the IgG analysis, DBS with glutathione-$S$-transferase (GST) MFI-bg reads above 500 (indicating non-specific binding) were excluded from the analysis. To dichotomize seropositivity to each species MSP1 antigen, a 2-component finite mixture model (FMM) of the $\log_{10}$-transformed MFI-bg signal was fit for each IgG response[37], and the positivity cutoff was determined by the lognormal mean of the first component (presumed seronegative) plus 2 standard deviations[42]. A 6-component model was created for PfMSP1 to more accurately define the seronegative population. FMM plots were generated in SAS version 9.4.

Remaining analyses were conducted in R version 4.1.1 (R Foundation for Statistical Computing, Vienna, Austria) or Microsoft® Excel® version 2208. Estimation of malaria exposure was adjusted while taking into account the complex sample design of the NAIIS using the R *survey* package[50], and data were weighted using normalized survey weights. Demographic, socioeconomic, and behavioral risk factors were considered in relation to malaria infection and exposure, including sex, age group, wealth quintile, and living in a household with at least 1 mosquito net per 1.8 household members. Since there was no data collected on mosquito net usage per child, a proxy net coverage variable was created using the ratio of nets to household members, based on the WHO criteria recommended for procurement purposes to ensure universal net coverage in households[51]. Cluster level covariates included: place of residence (urban/rural) and community net coverage. Community net coverage was defined as the proportion of individuals in the cluster with adequate household net coverage (1 net per 1.8 household members). Individual net coverage was normalized by subtracting the average community net coverage from the individual-level variable[52]. Three multivariate, mixed-effects logistic regression models were fit to determine factors associated with malaria exposure (measured as IgG positivity to PmMSP1, PoMSP1, or PvMSP1) and included a random intercept for the cluster variable. For the malaria infection risk factor analysis, multivariate logistic regression models were fit without random effects since there were only 1204 observations with over 800 clusters represented (average of 1.4 persons per cluster). Firth's penalized-likelihood logistic regression was performed for the *P. malariae* and *P. ovale* infection models to reduce the bias due to small sample size and quasi-complete and complete separation of the wealth covariate for the *P. malariae* and *P. ovale* models, respectively[53].

To model the rate of change from seronegative to seropositive for each of the three antigens (PmMSP1, PoMSP1, and PvMSP1), serocatalytic conversion models were fit to the seropositivity by age data for each antigen[54,55]. The model estimated the seroconversion rate (SCR or λ) which is the mean annual rate of conversion from any individual in the population from seronegative to seropositive, and the seroreversion rate (SRR or ρ) which is the mean annual rate of reversion from seropositive to seronegative. Children under age 1 year were excluded to remove the potential effect of maternally-derived antimalarial antibodies. The parameters were estimated using Markov Chain Monte Carlo (MCMC) with 10,000 MCMC iterations. R code to fit these models was utilized from the code provided by Dr. Michael White, Serology Github Repository[56]. The geospatial analysis was conducted in R version 4.1.1 and QGIS version 3.16.3, and spatially smoothed sero-prevalence maps were created using local Empirical Bayes spatial smoothing. All code is available at: https://github.com/cleonard297/nonPf_seroPCR_code.

## Reporting summary

Further information on research design is available in the Nature Portfolio Reporting Summary linked to this article.

## Data availability

DHS data are publicly available for download. NMS4 data are owned by the Government of Nigeria (GoN); requests for NMS4 data must be approved by GoN and all other NMS4 principal investigators. Inquiries should be directed to the corresponding author.

## Code availability

All code is available at: https://github.com/cleonard297/nonPf_seroPCR_code.

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

## Acknowledgements

The authors would like to acknowledge the field teams of the NAIIS survey staff as well as the Nigerian survey participants. The authors acknowledge the contribution of valuable specimens by the NAIIS Group, including the Federal Ministry of Health (FMoH), National Agency for the Control of AIDS (NACA), National Population Commission (NPopC), National Bureau of Statistics (NBS), U.S. Centers for Disease Control and Prevention (CDC), The Global Funds to Fight AIDS, Tuberculosis, and Malaria, University of Maryland Baltimore (UMB), ICF International, African Field Epidemiology Network (AFENET), University of Washington (UW), the Joint United Nations Programme on HIV and AIDS (UNAIDS), World Health Organization (WHO), and United Nations Children's Fund (UNICEF). The NMS4 project was financed by contributions by the President's Emergency Plan for AIDS Relief (PEPFAR), the US President's Malaria Initiative, the Global Immunization Division of the US Centers for Disease Control and Prevention, and The Global Fund to Fight AIDS, Tuberculosis, and Malaria, and the Bill and Melinda Gates Foundation through CDC Foundation. The findings and conclusions in this report are those of the authors and do not necessarily represent the official position of the US Centers for Disease Control and Prevention, US Agency for International Development, or NAIIS group and Nigeria Centre for Disease Control.

## Author contributions

P.U., U.I., A.O., S.M.G., M.I.O., and I.M. were involved with protocol formation and survey execution. S.M.G. and M.I.O. coordinated fieldwork for surveys. C.H., A.A., S.M.G., N.C.I., N.M., M.I.O., and E.R. coordinated laboratory work. D.M., L.S., and E.R. provided data management. M.M. and L.S. provided consultation and malaria expertize. C.M.L. and E.R. analyzed data. C.H., C.M.L., and E.R. drafted paper. All authors approved final version of the paper.

## Competing interests

The authors declare no competing interests.
