## [Peer Review File · Nature Communications]

Non-falciparum malaria infection and IgG seroprevalence among children under 15 years in Nigeria, 2018REVIEWER COMMENTS

Reviewer #1 (Remarks to the Author):

This is a large-scale cross-sectional study based on 31,234 dried blood spot samples from children, collected through a national Nigerian survey of HIV prevalence, that attempts to provide local, regional and national estimates of the infection prevalence and seroprevalence of the neglected malaria parasites *Plasmodium malariae*, *P. ovale* spp. and *P. vivax*. The use of both molecular methods (to evaluate current infection prevalence) and serology (to evaluate past exposure) is a major strength of the present study. Given the scarcity of data regarding the transmission of these neglected parasites this effort is commendable and the results would be of interest for researchers studying malaria and neglected tropical diseases as well as for malaria control programmes and other policy makers. However, I do have several concerns.

Major comments:

1)

My main concern is that the actual estimates of the current infection prevalence are based PCR analysis of a limited subset of the total number of samples (1204 out of 31234 i.e. approx. 3.9 %). This subset was selected by the authors using a two-stage selection strategy based on the presence of *Plasmodium* antigenaemia as determined by a multiplex immuno-assay (Strategy 1: low PfHRP2 and high pan-plasmodium antigenaemia; Strategy 2: High PfHRP2 antigenaemia).

It has been well described (e.g. reviewed by Sutherland Trends in Parasitology 2016) that both *P. malariae* and *P. ovale* spp. often cause low density infections, undetectable by microscopy and conventional antigen-detecting RDTs, that can only be detected using molecular methods. I am concerned that the current approach based on the presence of *Plasmodium* antigenaemia could provide a sample subset of "higher density" infections that is not representative of the population as a whole and could both over or underestimate the true burden of infection.

According to the authors, selection strategy 1 would "favour the identification of single species non-falciparum infections", however, it is evident from the results presented supplementary figure 3 that both selection strategies predominantly identify samples with a high likelihood of *P. falciparum* infection. The finding of a similar overall rate of PCR positivity of non-falciparum infections in the subsets of samples selected by both strategy 1 and strategy 2 could support that the estimates are truly representative however, it is not possible to evaluate this from the results presented within manuscript, particularly since samples with low levels of antigenaemia for all antigens does not appear to have been tested by PCR.

I worry that the selection strategy could produce biased estimates of infection prevalence and I would suggest that the authors perform a formal validation of the sample selection strategy to demonstrate whether the estimates of infection prevalence obtained are comparable to those that would have been obtained, should a larger sample set have been analysed directly by *plasmodium*-specific PCR.

Furthermore, it is not quite clear why so few samples were tested by PCR. I suspect time and cost were limiting factors but it would be good if this was evident within the manuscript text.

If the resources for PCR assays are limited and the expected prevalence of infection is relatively low (as it is for *P. malariae* and *P. ovale* spp.) an alternative approach that could have been considered by the authors would have been to pool samples (e.g. 10 samples at a time) and at a first stage run PCR on sample pools and then then at a second stage run PCRs on individual samples in pools that are positive for *P. malariae* and/or *P. ovale* spp.

2)

The methods section currently does not provide sufficient detail to fully evaluate the results or to replicate the results.

a) The bulk of the results are based on multiplex immunoassays used to assay >30,000 samples. There is currently no description of how batch-to-batch variation was accounted for in either the antigen detection assays or the serological assays. This is critical information for the interpretation of the results. Looking at the references cited by the authors regarding the serological assays (ref. 39 and 42; which appear to describe similar but not exactly the same method as used within the present study) it appears as high- and low-reactivity positive controls were used for batch correction but whether this was the case in the present study is not quite clear.

Please provide information regarding batch-to-batch-correction within the methods section detailing both the antigen detection assays and the serological assays. This should also include information on the technical controls used. Please also provide data on the magnitude of batch-to-batch and plate to plate variation.

b) The methods section regarding statistical and data analysis is very brief in describing both the complex multilevel regression models as well serocatalytic models. I would strongly urge the authors to submit the actual code used to analysed the present data either as supplementary material or preferably to deposit it within an appropriate repository. Furthermore, for the serocatalytic modelling of antibody prevalence data the authors simply refer to a github repository hosted by Dr. Michael White but states that the code has been modified but do not indicate how. Submitting the actual code together with the manuscript (preferably with some data to provide a minimal reproducible example) would give the interested reviewer/reader the opportunity to scrutinise the analysis.

Minor Comments:

This is a matter of personal preference. The results section is quite brief and given the Nature Communications article style with the methods section appearing at the end of the manuscript I think providing a little bit more detail regarding laboratory and data analysis within the results section itself would be helpful for the reader and give the manuscript a better flow.

Non standard abbreviations are not defined at the first occurrence within the text (sometimes not at all) which makes the manuscript quite difficult to read. Please correct this.

Lines 220-223: This sentence seems a bit out of place. Would fit better at the very beginning of the discussion.

Line 293: Please substitute "lifetime exposure" for "prior exposure"

Please specify the target genes of the PCR method within the methods section itself.

Antigen-level thresholds for sample selection strategies are not quite clear. The threshold for pLDH appears to be relative to the PfHRP2 level. How was this defined?

Regarding the thresholds of seropositivity. Why 6 component model for Pf. Was there a formal decision rule in selecting optimal number of components?

There is no comment within manuscript regarding potential cross-reactivity between orthologue Plasmodium proteins from different species. This is an important piece of information when interpreting the serological data. According to the findings by Priest et al. Mal. J 2018 cross-reactivity appears to be low, but this could preferably be stated somewhere in the manuscript as this strengthens the results and the conclusions presented by the authors.

Reference 53: The reference does not include a link to the appropriate github repository.

Reviewer #2 (Remarks to the Author):

The authors report parasite rates and IgG seroprevalence of human non-falciparum malaria parasites in children under 15 years of age, assessed in different geopolitical zones in Nigeria. More studies are needed to breach the knowledge gap of non-falciparum malaria parasites in endemic communities. The methods and data analysis are within acceptable standards but I have highlighted a few concerns below that the authors need to address to improve the quality of the manuscript.

The authors have used previously described methods for both DNA and antibody detection assays with the references appropriately cited. However, more details of the assays are required, for example, were the multiplex bead assays run in duplicates/triplicates, what controls were used and how was cross-reactivity between parasite species accounted for? There were several instances where the authors have mentioned significant differences from comparisons (e.g Page 7 line 164) but have not indicated a p-value so it is difficult to assess the basis of the significance reported.

The authors highlight a strong point of accessing samples from the different geopolitical zones in Nigeria, however this is not reflected in their discussion of either parasite rates or seroprevalence across the different geopolitical zones or even ecological or malaria transmission settings. Despite contributing approx. 27% of global malaria cases, malaria transmission is still very much heterogenic in Nigeria and the authors should have analyzed or discussed their results from that angle.

The authors acknowledge that the study design allowed them access to mostly asymptomatic to mildly symptomatic malaria cases, this is very critical and should be expounded on further as a lot of the interpretation of the results depend on this fact. Presence of parasites detected by PCR is not routine for description of active malaria infections so it will be more interesting if the authors are able to stratify their results, either by available information on symptoms or quantified parasite density. Also, the timeframe for sample collection from the different LGAs was not mentioned or taken into consideration in the analysis/discussions.

Reviewer #3 (Remarks to the Author):

This paper used data from a nationally representative HIV survey among children aged 0-14 in Nigeria to estimate infection and exposure to Plasmodium parasites. Overall, this is an impressive undertaking with a large sample size and important findings. However, I believe the clarity of the manuscript could be improved, particularly surrounding how the methods are explained and the results are presented. My comments and suggestions are in hopes of improving the manuscript.

Abstract

The abstract needs a sentence near the end highlighting why these findings are important/meaningful.

Introduction

Perhaps not in the introduction, but somewhere in this paper the authors should explain why they assayed children aged 0-14, given the survey was among all ages (correct?).

Results

A broad comment about the Results: this is up to the author's discretion, but I believe some of the results presented in supplements should be moved to the main manuscript. If the authors spend substantial time (more than 1-2 sentences) discussing a finding in the supplement, why not put it in the main paper? From a reader standpoint, it makes ease of reading much easier than flipping back and forth between supplements, paper, etc.

Line 83: What was the denominator of children aged < 15 surveyed during the NAIIS (i.e., what is the completeness of your data of children with antigen data)?

Line 84: Define LGA.

Line 90: Explain why the exact target of 100 was not met for each of the zones.

Line 97: Do the authors have any intuition as to why such a high proportion (27%) of those with now/low HRP2 were negative by Plasmodium genus primers?

Page 5: Supplemental Figure 3 is, I believe, a figure that could be shifted to the main paper. I personally find it much easier to digest the information in figure form. Regardless, please add percentages to the final boxes (percent Pf/Po, Pf/Pm, etc.) as you present them in the text.

Line 116: Was this supposed to reference supplemental figure 3? I'm not understanding the reference to supplemental figure 4.

Table 2: I'm a bit puzzled why the authors don't also have a model with Pf infection alone as an outcome. It would be useful to compare the results across models. If you don't want to add it, please justify the choice to only look at predictors of Pm and Po in the Methods.

Line 127 (sentence beginning "Significant associations were..."): What does this sentence indicate? Perhaps a broad summary of these findings would be useful. Also, please include the denominator for this analysis in the table (Supplemental table 2), as it's my understanding that this is among only those who tested positive for Plasmodium infection.

Figure 3: Please increase the axis fonts, very hard to read.

Serology results: Again, perhaps some explanation about why the authors didn't perform analyses/present results from catalytic models for PfMSP1 would be nice.

Supplementary Table 4: Why is there no equivalent to this table (bivariate analysis) for the Plasmodium infection findings?

Page 8 (subheading 'Combination of PET-PCR and serological data'): These findings would warrant some explanation (perhaps in the discussion). What do they mean?

Discussion

Personal preference, but I think the final sentence of the first paragraph of the discussion would be a nice first sentence of the Discussion as it summarizes what the authors have done in this study.

Line 275: The authors mention that having larger standard errors should be interpreted with caution – perhaps the map of SEs should go in the main paper? Unsurprisingly, the uncertainty for vivax and ovale is really high, so it's good to caution readers about interpretation.

Methods

Page 15: The description of the survey is nice, but please add in a sentence or two explaining how you've selected your final sample (children aged <15). Were all children in selected households sampled, or just a sub-sample?

Page 17, line 406: Can the authors provide a bit of justification for the 600 DBS sample size for the second selection criteria? Was there some sort of sample size calculation justification for this?

Reviewer #1 (Remarks to the Author):

This is a large-scale cross-sectional study based on 31,234 dried blood spot samples from children, collected through a national Nigerian survey of HIV prevalence, that attempts to provide local, regional and national estimates of the infection prevalence and seroprevalence of the neglected malaria parasites *Plasmodium malariae*, *P. ovale* spp. and *P. vivax*. The use of both molecular methods (to evaluate current infection prevalence) and serology (to evaluate past exposure) is a major strength of the present study. Given the scarcity of data regarding the transmission of these neglected parasites this effort is commendable and the results would be of interest for researchers studying malaria and neglected tropical diseases as well as for malaria control programmes and other policy makers. However, I do have several concerns.

Major comments:

1) My main concern is that the actual estimates of the current infection prevalence are based PCR analysis of a limited subset of the total number of samples (1204 out of 31234 i.e. approx. 3.9 %). This subset was selected by the authors using a two-stage selection strategy based on the presence of *Plasmodium* antigenaemia as determined by a multiplex immuno-assay (Strategy 1: low PfHRP2 and high pan-plasmodium antigenaemia; Strategy 2: High PfHRP2 antigenaemia).

It has been well described (e.g. reviewed by Sutherland Trends in Parasitology 2016) that both *P. malariae* and *P. ovale* spp. often cause low density infections, undetectable by microscopy and conventional antigen-detecting RDTs, that can only be detected using molecular methods. I am concerned that the current approach based on the presence *Plasmodium* antigenaemia could provide a sample subset of “higher density” infections that is not representative of the population as a whole and could both over or underestimate the true burden of infection.

According to the authors, selection strategy 1 would “favour the identification of single species non-falciparum infections”, however, it is evident from the results presented supplementary figure 3 that both selection strategies predominantly identify samples with a high likelihood of *P. falciparum* infection. The finding of a similar overall rate of PCR positivity of non-falciparum infections in the subsets of samples selected by both strategy 1 and strategy 2 could support that the estimates are truly representative however, it is not possible to evaluate this from the results presented within manuscript, particularly since samples with low levels of antigenaemia for all antigens does not appear to have been tested by PCR.

I worry that the selection strategy could produce biased estimates of infection prevalence and I would suggest that the authors perform a formal validation of the sample selection strategy to demonstrate whether the estimates of infection prevalence obtained are comparable to those that would have been obtained, should a larger sample set have been analysed directly by *Plasmodium*-specific PCR.

- **The authors thank the reviewer for raising this important point regarding the PCR confirmation for *Plasmodium* species in this study. We do agree that both *P. malariae* and *P. ovale* are many times found as “low-density” infections that would be missed by traditional diagnostics of microscopy and RDTs (which are developed for healthcare settings). An**

important difference in our study is that the first screen for active infection was performed with a lab-based multiplex antigen assay which has a level of detection well below that of RDTs. To the reviewer's point, it is important to note that selection strategy 1 required samples to be positive to the non-HRP2 target to be selected, so even if the samples had low levels of confirmed antigenemia, these were still eligible for selection. If low (but positive) levels for multiple antigen targets would be observed, this sample would be selected as the HRP2 level is expected to be elevated in relation to the other targets in a "normal" Pf infection (shown by scatterplots in Supp Fig 2).

But the reviewer also raises an important point: what is the PCR-determined sensitivity of our strategy if a blood sample would be found negative to all antigen targets? To address the question for this particular Nigeria sample set, we have gone back and selected 200 random DBS from this survey that were antigen negative for all targets, extracted DNA, and performed our same PET-PCR speciation assays. Of all 200 DBS, zero were found to have *P. ovale* or *P. vivax* DNA, but 3 (1.5%) were positive for *P. malariae* DNA. In comparison to the 6.6% estimate of active *P. malariae* infections from our study, this 1.5% is not a negligible finding, but we feel would not influence our overall findings for risk factor analyses and geospatial findings based on active infection status. We feel that our pragmatic approach (further emphasized below) for selection of specimens for species PCR was appropriate given the magnitude of blood specimens available and the outputs we present in this report. We have explained this further validation and discovery of the additional Pm infections among antigen negative DBS in the "Sample Selection" section of the Methods, and have also included additional text in the limitations section of Discussion that certainly low-density non-Pf infections were missed in this study.

Furthermore, it is not quite clear why so few samples were tested by PCR. I suspect time and cost were limiting factors but it would be good if this was evident within the manuscript text.

If the resources for PCR assays are limited and the expected prevalence of infection is relatively low (as it is for *P. malariae* and *P. ovale* spp.) an alternative approach that could have been considered by the authors would have been to pool samples (e.g. 10 samples at a time) and at a first stage run PCR on sample pools and then then at a second stage run PCRs on individual samples in pools that are positive for *P. malariae* and/or *P. ovale* spp.

- **Following the information provided above regarding the sensitivity of the antigen detection assay, the authors wanted to take a realistic approach to specimens selected to undergo DNA extraction and PCR identification. With the >31k samples included in this study, PCR would have been an incredible and expensive task for all of these specimens. Even if pooling by multiples of 10, this would also have reduced the sensitivity of the PCR assay, and as the reviewer notes, would ultimately also require disaggregation of the pools in order to have individual child data. Per the additional PCRs performed on antigen negative samples described above, the authors have further supplemented the Methods and limitations section the rationale for sample selection for PCRs and how a small proportion of non-Pf infections will inevitably be missed.**

2) The methods section currently does not provide sufficient detail to fully evaluate the results or to replicate the results.

a) The bulk of the results are based on multiplex immunoassays used to assay >30,000 samples. There is currently no description of how batch-to-batch variation was accounted for in either the antigen detection assays or the serological assays. This is critical information for the interpretation of the results. Looking at the references cited by the authors regarding the serological assays (ref. 39 and 42; which appear to describe similar but not exactly the same method as used within the present study) it appears as high- and low-reactivity positive controls were used for batch correction but whether this was the case in the present study is not quite clear.

Please provide information regarding batch-to-batch-correction within the methods section detailing both the antigen detection assays and the serological assays. This should also include information on the technical controls used. Please also provide data on the magnitude of batch-to-batch and plate to plate variation.

- The authors thank the reviewer for this suggestion and apologize for not including more information about assessing plate variation for the antigen detection and IgG detection assays. For both types of immunoassays, a pass/fail scheme dependent upon controls included on every assay plate was employed for accepting/rejecting the plate data for the unknowns.

For the antigen detection assay, this further detail has now been added within the appropriate sub-section of the Methods and is referenced by the previous quality assurance methodology cited (Alvarado, et al). In total, 1.2% of all antigen detection plates initially failed and needed to be repeated, and this is now stated in Results.

For the IgG detection assay, positive/negative controls were included on each assay plate, and a non-binding internal control bead (GST) was also included in each assay well in a similar manner to previous work from our group which is now referenced here (van den Hoogen, et al). As currently stated in Results, 0.72% of all samples were excluded from analyses for evidence of GST (non-specific) binding. The pass/fail scheme for IgG detection plates required a negative signal for the assay plate malaria IgG negative control as well as the malaria IgG positive control signal to be within 2sd of the moving average. This information has now been added to the section in Methods describing the multiplex IgG assay.

b) The methods section regarding statistical and data analysis is very brief in describing both the complex multilevel regression models as well serocatalytic models. I would strongly urge the authors to submit the actual code used to analysed the present data either as supplementary material or preferably to deposit it within an appropriate repository. Furthermore, for the sero-catalytic modelling of antibody prevalence data the authors simply refer to a github repository hosted by Dr. Michael White but states that the code has been modified but do not indicate how. Submitting the actual code together with the manuscript (preferably with some data to provide a minimal reproducible example) would give the interested reviewer/reader the opportunity to scrutinise the analysis.

- **We thank the reviewer for this suggestion and are in complete agreement that this code utilized for this study should be deposited. We have now uploaded this to GitHub and included the link at the end of the Statistical Analysis section, as well as the Code Availability statement.**

There was a poor choice of wording on the authors' part in describing the code initially provided by Dr. White in that our group didn't "adapt" (i.e. modify/adjust) the fundamental code itself, but just had changed the variables to match those in our data. We have revised this sentence to read: "R code to fit these models was utilized from the code provided by Dr. Michael White, Serology Github Repository".

Minor Comments:

This is a matter of personal preference. The results section is quite brief and given the Nature Communications article style with the methods section appearing at the end of the manuscript I think providing a little bit more detail regarding laboratory and data analysis within the results section itself would be helpful for the reader and give the manuscript a better flow.

- **The authors thank the reviewer for this suggestion, and agree that given the journal style, more room is available to expand the explanation of results. We have gone throughout this section and provided more text for the readers' context.**

Non standard abbreviations are not defined at the first occurrence within the text (sometimes not at all) which makes the manuscript quite difficult to read. Please correct this.

- **The authors have reviewed the entire manuscript and have corrected these abbreviations at first mention.**

Lines 220-223: This sentence seems a bit out of place. Would fit better at the very beginning of the discussion.

- **The authors agree with this suggestion, and have moved this sentence to the beginning of Discussion.**

Line 293: Please substitute "lifetime exposure" for "prior exposure"

- **This has been changed.**

Please specify the target genes of the PCR method within the methods section itself.

- **We have added this specific information for all of the primer targets used in this study, and appropriate references.**

Antigen-level thresholds for sample selection strategies are not quite clear. The threshold for pLDH appears to be relative to the PfHRP2 level. How was this defined?

- **Yes, the reviewer is correct in stating that the first selection strategy is contingent on the relative assay signal of HRP2 antigen as compared with the other antigen targets. This strategy relied on visual identification of the scatterplot of assay signals of HRP2 versus other targets (displayed in Supp Fig 2), and observing which specimens appeared to have phenotypic**

evidence of non-Pf infection. Additional text and references have been added to this section of Methods to further explain. We feel that this selection strategy in conjunction with the second strategy to select for DBS with high HRP2 levels (known Pf infections) provided a thorough methodology to find both single-species non-Pf infections as well as those non-Pf species mixed with Pf parasites.

Regarding the thresholds of seropositivity. Why 6 component model for Pf. Was there a formal decision rule in selecting optimal number of components?

- **Due to the negative skewness of the Pf IgG data (as expected in this high Pf endemic setting), the two-component model did not fit well in parametrizing the leftmost, putative seronegative component for PfMSP1. Successive model fittings of 3, 4, and 5 components each improved this fitting, but it wasn't until fitting this 6-component model that this actual leftmost component was clearly visualized. This was not a problem for the positively-skewed PmMSP1, PoMSP1, and PvMSP1 IgG responses (Supp Fig 6).**

There is no comment within manuscript regarding potential cross-reactivity between orthologue Plasmodium proteins from different species. This is an important piece of information when interpreting the serological data. According to the findings by Priest et al. Mal. J 2018 cross-reactivity appears to be low, but this could preferably be stated somewhere in the manuscript as this strengthens the results and the conclusions presented by the authors.

- **The authors thank the reviewer for this suggestion, and agree that this point needs to be emphasized for this study. As currently stated in Results, we did assess IgG cross-binding among all of the MSP1-19kD orthologues as shown in Supp Fig 2, and found no correlation in the magnitude of IgG signal for any one antigen compared to another within this study population. We have added a sentence to Discussion to reiterate the previous Priest, et al, findings with these four MSP1-19kD antigens, as well as our current findings in this Nigerian study population which appears to confirm no appreciable IgG cross-binding.**

Reference 53: The reference does not include a link to the appropriate github repository.

- **Regarding the major comment 2b) above, we have included the github link inclusive of all R code utilized in this study.**

Reviewer #2 (Remarks to the Author):

The authors report parasite rates and IgG seroprevalence of human non-falciparum malaria parasites in children under 15 years of age, assessed in different geopolitical zones in Nigeria. More studies are needed to breach the knowledge gap of non-falciparum malaria parasites in endemic communities. The methods and data analysis are within acceptable standards but I have highlighted a few concerns below that the authors need to address to improve the quality of the manuscript.

The authors have used previously described methods for both DNA and antibody detection assays with the references appropriately cited. However, more details of the assays are required, for example, were the multiplex bead assays run in duplicates/triplicates, what controls were used and how was cross-reactivity between parasite species accounted for?

- **The authors thank the reviewer for pointing this out, and we have added more text to the respective sections of Methods to better explain our assays and quality control procedures. To the reviewer's specific questions above, the multiplex bead assays were all run in singlet (due to the number of specimens), known positive and negative controls were included on each assay plate, and cross-reactivity was assessed as shown in Supplemental Figure 2 and had shown no significant evidence of cross-binding.**

There were several instances where the authors have mentioned significant differences from comparisons (e.g Page 7 line 164) but have not indicated a p-value so it is difficult to assess the basis of the significance reported.

- **The authors appreciate this comment, but would point to the 95% confidence intervals included on many of the figures which allows for assessment of statistical significance. For example, the significant differences found among the seroconversion rate (SCR) estimates among the PmMSP1, PoMSP1, and PvMSP1 antigens is displayed by the non-overlapping confidence intervals for all three of these antigens in Figure 3a.**

The authors highlight a strong point of accessing samples from the different geopolitical zones in Nigeria, however this is not reflected in their discussion of either parasite rates or seroprevalence across the different geopolitical zones or even ecological or malaria transmission settings. Despite contributing approx. 27% of global malaria cases, malaria transmission is still very much heterogenic in Nigeria and the authors should have analyzed or discussed their results from that angle.

- **We are in agreement with the reviewer's comment that the malaria exposure in Nigeria resulting from infection with any of the human *Plasmodium* spp. appears to be heterogeneous. For children with active parasite infection, we display these results by Nigerian state in Figure 2. For the serological data for which more 'positives' were available, we present the state-level estimates in Supp Table 3 and the spatial estimates at the lowest administrative level possible in Figure 3 – the local government area (LGA). However, to the reviewer's point, we have now added additional text to the end of the first paragraph of Discussion to emphasize that these non-falciparum species appear to have very heterogeneous transmission throughout Nigeria – which is in the same non-homogenous manner as *P. falciparum*, though not in the same areas in the country.**

The authors acknowledge that the study design allowed them access to mostly asymptomatic to mildly symptomatic malaria cases, this is very critical and should be expounded on further as a lot of the interpretation of the results depend on this fact. Presence of parasites detected by PCR is not routine for description of active malaria infections so it will be more interesting if the authors are able to stratify their results, either by available information on symptoms or quantified parasite density. Also, the timeframe for sample collection from the different LGAs was not mentioned or taken into consideration in the analysis/discussions.

- **Unfortunately, participant symptomatic status indicative of malaria (febrile, chills, etc.) was not captured by this 2018 HIV survey, so that data is not available for analyses. All we really know is that the participants were not treatment-seeking at the time of enrollment at their households, and we have expounded on this fact in Discussion. Participants were enrolled in the survey from July-December of 2018 throughout the different LGAs, so there's no capacity to structure analyses on time of sample collection.**

Reviewer #3 (Remarks to the Author):

This paper used data from a nationally representative HIV survey among children aged 0-14 in Nigeria to estimate infection and exposure to Plasmodium parasites. Overall, this is an impressive undertaking with a large sample size and important findings. However, I believe the clarity of the manuscript could be improved, particularly surrounding how the methods are explained and the results are presented. My comments and suggestions are in hopes of improving the manuscript.

Abstract

The abstract needs a sentence near the end highlighting why these findings are important/meaningful.

- **We thank the reviewer for this suggestion, but due to the limited word limit of the journal style, we are unable to supplement with additional text. We feel the current ending of the Abstract alludes to the importance of this work: "Serological and DNA indicators show widespread exposure of Nigerian children to Pm with lower rates to Po and Pv."**

Introduction

Perhaps not in the introduction, but somewhere in this paper the authors should explain why they assayed children aged 0-14, given the survey was among all ages (correct?).

- **The reviewer raises an important point here, and is correct that the 2018 HIV survey did collect specimens for persons of all ages. Ultimately, the focus on the children's data in this report was due to their samples being processed for IgG and antigen data collection first, and malaria susceptibility highest among children. As of late 2022, adult samples (~150,000 total) are still having malaria IgG and antigen data collected with the hope of future analyses inclusive of all participants. We have added this information to the "Laboratory data collection for malaria biomarkers" section of Methods to explain to the reader why only children's data are included here.**

Results

A broad comment about the Results: this is up to the author's discretion, but I believe some of the results presented in supplements should be moved to the main manuscript. If the authors spend substantial time (more than 1-2 sentences) discussing a finding in the supplement, why not put it in the main paper? From a reader standpoint, it makes ease of reading much easier than flipping back and forth between supplements, paper, etc.

- **The authors are in complete agreement with the reviewer's assessment, and given the formatting style of the journal, have seen opportunities to move material from Supplemental to the main body of Results. We have done this for Supplemental Figure 3 (now Figure 1), Supplemental Figure 8 (now Figure 4), and Supplemental Figure 13 (now Figure 6).**

Line 83: What was the denominator of children aged < 15 surveyed during the NAIS (i.e., what is the completeness of your data of children with antigen data)?

- **Of 45,462 eligible children (<15y) eligible for enrollment in NAIS, multiplex data was able to be collected on 31,234 (68.7%). This information has now been added to the first sentence of Results.**

Line 84: Define LGA.

- **This has now been defined.**

Line 90: Explain why the exact target of 100 was not met for each of the zones.

- **This text was an error on our part, and of the 600 samples selected (100 from each zone), a total of 4 were missing for further PCR analyses. This has now been corrected in the text and Figure 1.**

Line 97: Do the authors have any intuition as to why such a high proportion (27%) of those with now/low HRP2 were negative by Plasmodium genus primers?

- **These samples showing positive antigenemia but negative for *Plasmodium* DNA are likely very low density infections, or potentially infections that have been recently cleared of active parasitemia. As the laboratory antigen assay is very sensitive, it is likely these specimens simply had enough antigen in the whole blood to be detected by the immunoassay, but not enough DNA to be captured by the PCR.**

Page 5: Supplemental Figure 3 is, I believe, a figure that could be shifted to the main paper. I personally find it much easier to digest the information in figure form. Regardless, please add percentages to the final boxes (percent Pf/Po, Pf/Pm, etc.) as you present them in the text.

- **We are in agreement with the reviewer's suggestion, and have move Supp Fig 3 (among others) to the main body. We have also added percentages to the terminal boxes.**

Line 116: Was this supposed to reference supplemental figure 3? I'm not understanding the reference to supplemental figure 4.

- **The authors note the confusing reference here, and have now revised to have these findings refer to Supp Fig 3 (now Fig 1).**

Table 2: I'm a bit puzzled why the authors don't also have a model with Pf infection alone as an outcome. It would be useful to compare the results across models. If you don't want to add it, please justify the choice to only look at predictors of Pm and Po in the Methods.

- **The authors thank the reviewer for bringing up this point, but the emphasis of this current study was on non-falciparum infections and exposure in Nigeria. With the fundamental differences in Pf transmission and endemicity in Nigeria, as currently stated in the Discussion, the Pf data will be comprehensively presented in a forthcoming analytical study.**

Line 127 (sentence beginning "Significant associations were..."): What does this sentence indicate? Perhaps a broad summary of these findings would be useful. Also, please include the denominator for this analysis in the table (Supplemental table 2), as it's my understanding that this is among only those who tested positive for Plasmodium infection.

- **Initially, this sentence was just to indicate that versus Pf single-species infections, that infections containing Pm or Po showed additional significant associations, but per the reviewer's suggestion, we have expanded this text further to more clearly describe this analysis and findings included in Supplemental Table 2. In the figure legend, we have also indicated the sub-population included in the analysis here: "Among children with any *Plasmodium* infection..."**

Figure 3: Please increase the axis fonts, very hard to read.

- **These have been increased.**

Serology results: Again, perhaps some explanation about why the authors didn't perform analyses/present results from catalytic models for PfMSP1 would be nice.

- **The authors thank the reviewer for bringing this point up, but wish to focus on the non-falciparum findings for this particular report. We believe the map of PfMSP1 seropositivity shown in Fig 3 accentuates the ubiquitous transmission that we wanted to show for *P. falciparum* versus all other human malaria species. A future report will explain serocatalytic results for *P. falciparum* among many other metrics of transmission.**

Supplementary Table 4: Why is there no equivalent to this table (bivariate analysis) for the Plasmodium infection findings?

- **Unlike the more "standard" demographic and individual characteristics presented in the adjusted active infection analysis in Table 2, these factors included in the bivariate analysis here were subjected to a greater degree of missingness. Given the very low prevalence of active infections, we felt bivariate modelling for non-Pf exposure through IgG seropositivity would be the most appropriate strategy here.**

Page 8 (subheading 'Combination of PET-PCR and serological data'): These findings would warrant some explanation (perhaps in the discussion). What do they mean?

- **The authors thank the reviewer for this comment, and agree that we can expand on the description of these findings. We have added additional text to this section of Results as well**

as additional text in Discussion explaining higher IgG levels during active infection, and the concordance of sero and PCR results.

Discussion

Personal preference, but I think the final sentence of the first paragraph of the discussion would be a nice first sentence of the Discussion as it summarizes what the authors have done in this study.

- **The authors agree with the reviewer's suggestion, and have moved this sentence accordingly.**

Line 275: The authors mention that having larger standard errors should be interpreted with caution – perhaps the map of SEs should go in the main paper? Unsurprisingly, the uncertainty for vivax and ovale is really high, so it's good to caution readers about interpretation.

- **The authors feel the map of relative standard errors by LGA is truly supplemental information, but to the reviewer's point, we have added an additional sentence to the Discussion about RSEs to emphasize that lower-seroprevalence Po and Pv estimates are more prone to higher RSEs and should be interpreted accordingly. " Specifically, the overall lower IgG seroprevalence to the P. ovale and P. vivax MSP1 targets led to higher RSEs by LGA with most exceeding an RSE of 30%."**

Methods

Page 15: The description of the survey is nice, but please add in a sentence or two explaining how you've selected your final sample (children aged <15). Were all children in selected households sampled, or just a sub-sample?

- **All children providing a blood sample for the NAIS survey were included in this current study. We have added text to clarify this.**

Page 17, line 406: Can the authors provide a bit of justification for the 600 DBS sample size for the second selection criteria? Was there some sort of sample size calculation justification for this?

- **The selection of 100 Pf+ specimens from each of the six zones was designed to be geographically representative for P. falciparum infections throughout the entire country, and additional text has been added here to clarify this.**

REVIEWERS' COMMENTS

Reviewer #1 (Remarks to the Author):

I appreciate the opportunity to re-review the revised version of the manuscript and I believe the authors have now addressed most of the issues indicated in the initial reviewer report, however, I have a few remaining comments.

It is great that the authors have provided an additional evaluation of the selection strategy by running PET-PCR for 200 randomly selected samples negative for all antigenic targets. However, this data should be presented within the results section. The authors have not indicated how many of these samples (if any) that were positive for *P. falciparum*. To provide context for this evaluation of the performance of selection strategy 1 (which the authors use to identify samples with a higher likelihood of non-*falciparum* infection) I would suggest that the authors present PET-PCR data (including data on *P. falciparum*) for the 200 antigen negative samples together with data for 200 randomly selected samples out of 596 samples that selected by "strategy 1" (i.e. pan-*Plasmodium*-antigen positive but HRP2 low/negative). This could be presented in a table and included as supplementary information.

Furthermore, I would still urge the authors to provide a brief description of the sample selection strategy for PET-PCR-testing within the results section itself.

Very minor comments:

Line 85: Please define the abbreviation of "dried blood spots (DBS)" at first occurrence.

Line 97: What do the authors mean by "low/absent HRP2 present"? I suspect the authors mean low / undetectable levels of HRP2 but please revise for clarity.

Lines 144-149: For this reviewer these two sentences are quite difficult to follow. Does this refer to mixed species infections? Consider rephrasing for clarity.

Line 428: Please define "buffer background (bg)"

Reference 56: This reference is still missing the URL for the github repository it is referring to. Please add.

Supplementary Fig. 9: Please clarify in figure legend and plot titles that the data presented are serological data on species specific IgG antibody responses towards MSP1.

Reviewer #2 (Remarks to the Author):

I am satisfied with the authors' responses and improvements made to the manuscript and I therefore recommend publication. Thank you

NCOMMS-22-32765A

Response to Reviewers

Reviewer #1 (Remarks to the Author):

I appreciate the opportunity to re-review the revised version of the manuscript and I believe the authors have now addressed most of the issues indicated in the initial reviewer report, however, I have a few remaining comments.

It is great that the authors have provided an additional evaluation of the selection strategy by running PET-PCR for 200 randomly selected samples negative for all antigenic targets. However, this data should be presented within the results section. The authors have not indicated how many of these samples (if any) that were positive for *P. falciparum*. To provide context for this evaluation of the performance of selection strategy 1 (which the authors use to identify samples with a higher likelihood of non-*falciparum* infection) I would suggest that the authors present PET-PCR data (including data on *P. falciparum*) for the 200 antigen negative samples together with data for 200 randomly selected samples out of 596 samples that selected by "strategy 1" (i.e. pan-*Plasmodium*-antigen positive but HRP2 low/negative). This could be presented in a table and included as supplementary information.

We thank the reviewer for the positive feedback and the further suggestions. We agree the text for PCR results for this panel of 200 antigen negatives should be included within the main Results, and have now moved this text from Methods to line 111 of Results. We have also included here the number of antigen negative samples positive for *P. falciparum* DNA: 9, 4.5%. However, as we already provide the exact numbers and percentages for DNA positives separately for samples selected by strategies 1 and 2, we do not feel that a comparison with a hypothetical 200 randomly selected samples from strategy 1 is warranted.

Furthermore, I would still urge the authors to provide a brief description of the sample selection strategy for PET-PCR-testing within the results section itself.

We thank the reviewer for this suggestion, and have included supplemental text within this paragraph of results beginning line 98 to more clearly describe where how the samples for PCR analysis were selected.

Very minor comments:

Line 85: Please define the abbreviation of "dried blood spots (DBS)" at first occurrence.

The has now been done.

Line 97: What do the authors mean by "low/absent HRP2 present"? I suspect the authors mean low / undetectable levels of HRP2 but please revise for clarity.

Yes, this is correct. We have revised this to "low / undetectable levels of HRP2"

Lines 144-149: For this reviewer these two sentences are quite difficult to follow. Does this refer to mixed species infections? Consider rephrasing for clarity.

We have revised this first sentence to hopefully read clearer. “Among children with any Plasmodium infection, some significant associations were also observed comparing infections with P. falciparum alone versus a mixed (or mono) infection containing P. malariae or P. ovale (or both)(Supplementary Table 2).”

Line 428: Please define “buffer background (bg)”

This has now been added here: “the MFI signal for sample dilution buffer (buffer background)”

Reference 56: This reference is still missing the URL for the github repository it is referring to. Please add.

We apologize for the omission, and this has now been added.

Supplementary Fig. 9: Please clarify in figure legend and plot titles that the data presented are serological data on species specific IgG antibody responses towards MSP1.

This has now been done.

Reviewer #2 (Remarks to the Author):

I am satisfied with the authors' responses and improvements made to the manuscript and I therefore recommend publication. Thank you

We thank the reviewer for the positive comments.